# Alleviating Effect of α-Lipoic Acid and Magnesium on Cadmium-Induced Inflammatory Processes, Oxidative Stress and Bone Metabolism Disorders in Wistar Rats

**DOI:** 10.3390/ijerph16224483

**Published:** 2019-11-14

**Authors:** Iwona Markiewicz-Górka, Krystyna Pawlas, Aleksandra Jaremków, Lidia Januszewska, Paweł Pawłowski, Natalia Pawlas

**Affiliations:** 1Department of Hygiene, Wroclaw Medical University, Mikulicza-Radeckiego 7, Wroclaw, 50-345, Poland; krystyna.pawlas@umed.wroc.pl (K.P.); aleksandra.jaremkow@umed.wroc.pl (A.J.); lidia.januszewska@umed.wroc.pl (L.J.); 2Specialist Hospital dr Alfred Sokołowski, Sokołowskiego 4, Wałbrzych 58-309, Poland; paweltoxic@tlen.pl; 3Department of Pharmacology, Faculty of Medical Sciences in Zabrze, Medical University of Silesia, Jordana 38, Zabrze 41-808, Poland; n-pawlas@wp.pl

**Keywords:** cadmium, magnesium, α-lipoic acid, oxidative stress, C-reactive protein, rats

## Abstract

Cadmium exposure contributes to internal organ dysfunction and the development of chronic diseases. The aim of the study was to assess the alleviating effect of α-lipoic acid and/or magnesium on cadmium-induced oxidative stress and disorders in bone metabolism, kidney and liver function, and hematological and biochemical parameters changes. Male rats were exposed to cadmium (30 mg Cd/kg of feed) for three months. Some animals exposed to Cd were supplemented with magnesium (150 mg Mg/kg of feed) and/or with α-lipoic acid (100 mg/kg body weight, four times a week). Cd intake inhibited body weight gain and lowered hemoglobin concentration, whereas it increased the activities of liver enzymes, as well as the level of oxidative stress, CTX-1 (C-terminal telopeptide of type I collagen, bone resorption marker), and CRP (C-reactive protein, marker of inflammation); it decreased vitamin D3, GSH (reduced glutathione), and the serum urea nitrogen/creatinine index. Mg and/or α-lipoic acid supplementation increased the antioxidant potential, and partially normalized the studied biochemical parameters. The obtained results show that both magnesium and α-lipoic acid decrease oxidative stress and the level of inflammatory marker, as well as normalize bone metabolism and liver and kidney function. Combined intake of α-lipoic acid and magnesium results in reinforcement of the protective effect; especially, it increases antioxidant defense.

## 1. Introduction

Cadmium (Cd) is characterized by very high toxicity and, at the same time, remains one of the most common metals in the environment. It is present in the air, water, and food. The unique physicochemical properties of cadmium and its compounds (including resistance to corrosion and chemicals, low melting point, and excellent electrical conductivity) make it applicable in many areas of the economy [1]. Despite limitations and attempts to replace cadmium with other components, its use in industry is constantly increasing. It is applied, among others, in new technologies such as nanomaterials and photovoltaic cells [2]. Cd can be expected to be one of the most serious environmental threats to human health for many years in the future.

Professional exposure to cadmium involves, among others, workers of non-ferrous metal smelters hut; welders; and people employed in the production of nickel-cadmium batteries, alloys, anticorrosive coatings, photovoltaic cells (solar panels), pesticides, plastics, and cadmium pigments. For the general population, food, smoking, and exhaust fumes are the main sources of cadmium [3]. Cadmium exposure results in oxidative stress and inflammatory processes, and contributes to internal organ dysfunction and the development of chronic diseases (osteoporosis, cardiovascular diseases, neoplasms) [3,4,5]. In addition to proper cadmium waste management or effective solutions limiting the exposure of workers and the general population, cadmium toxicity alleviation by strengthening the organism from the inside may be an important element of health prophylaxis.

The induction of oxidative stress by cadmium, which is not a transition metal, is explained by various mechanisms; for example, displacement by Cd of Fenton metals (such as iron) from proteins, leading to the production of a toxic hydroxyl radical. Cd binding to the mitochondrial membrane causes oxidative phosphorylation disturbance, and generation of peroxide radical. As a result of sulfhydryl (SH) groups binding, cadmium causes depletion of reduced glutathione (GSH) resources and lowered antioxidant enzyme activities [4,5,6].

The concept of cadmium toxicity alleviation and disease development inhibition by means of antioxidant or anti-inflammatory substances or elements acting antagonistically to cadmium is widely accepted. Many researchers are looking for effective supplements or their mixtures that could prevent or even reverse the pathological changes induced in tissues by cadmium. In a study by Imafidon et al. [7], treating rats with cadmium nephropathy with an extract from *Vernonia amygdalina*, a plant rich in antioxidant polyphenols, restored the proper functioning of kidneys, normalized urea and creatinine levels, increased the concentration of reduced glutathione, and decreased the level of malonylodialdehyde. Administration of selenium to layer hens decreased cadmium concentration in bones, stimulated the bone formation processes, and inhibited bone resorption [8]. In rats, intraperitoneal administration of strawberry (*Fragaria ananassa*) extract, containing flavonoids and polyphenols of antioxidant, anti-apoptotic, anti-inflammatory, and chelating impact, alleviated neurotoxic effects of cadmium, decreased Cd concentration in the brain, and reduced lipid peroxides level and the expression of proinflammatory cytokine TNF-α (tumor necrosis factor α) protein. At the same time, it increased GSH content and the activity of antioxidant enzymes, restoring redox balance in the brain tissue and partially improving the structure of damaged neurons [9].

In our experiment, we studied the protective effect of α-lipoic acid and magnesium against cadmium toxicity.

Magnesium and cadmium compete with each other on the level of absorption, binding with proteins transporting metals in the gastrointestinal tract, and accumulation in the tissues. Magnesium is an antioxidant agent, necessary for the de novo synthesis of glutathione. Mg ions react with phosphates and carboxyl groups of various components of cell membranes, affecting their stabilization, fluidity, and permeability, as well as the proper functioning of ion channels. Increased concentrations of inorganic magnesium compounds in bones raise their fracture resistance [10,11].

In turn, α-lipoic acid is a naturally occurring compound, synthesized in small amounts in animal and plant organisms. Significant sources of α-lipoic acid in the diet include spinach, broccoli, tomatoes, and liver. It is a saturated fatty acid in which three carbon atoms together with two sulfur atoms form a dithiolane ring. It is a strong antioxidant of both water and lipid phases, regenerates other antioxidants (e.g., vitamin E, vitamin C, glutathione), and is able to chelate heavy metals and reduce their toxicity [12,13].

The properties of α-lipoic acid and magnesium suggest that the mechanisms of their action may complement each other and enhance the body’s defense against cadmium-induced damage. However, the literature lacks documented data on this subject. Nevertheless, magnesium and α-lipoic acid are combined in some multicomponent dietary supplements. It is thus necessary to evaluate if the intake of such preparations is justified in individuals exposed to cadmium and if they can alleviate the toxic effect of the metal on the human body, as well as inhibit the pathological changes in internal organs and the development of chronic diseases.

Our research was conducted on an animal model (male rats). Its purpose was to assess the alleviating effect of α-lipoic acid and/or magnesium on cadmium-induced oxidative stress, inflammatory processes, changes in hematological and lipid parameters, as well as disorders in bone metabolism and kidney and liver function.

## 2. Materials and Methods

### 2.1. Chemicals

Cadmium chloride hemipentahydratae (CdCl_2_x2.5H_2_0) was obtained from Chempur (Piekary Śląskie, Poland). (R)-alfa lipoic acid 98% and magnesium glycerophosphate hydrate (C_3_H_7_MgO_6_P·xH_2_O) powder were obtained from Abcr (Karlsruhe, Germany). All reagents were of analytical grade.

### 2.2. Animals and Experimental Design

Thirty Wistar male rats (6–7 weeks old), initially weighing 200 (±27) g (mean ± SD), were used in this study. The animals were bred and kept in the Animal Laboratory of Wroclaw Medical University, Poland. The rats were housed in plastic cages under standard laboratory conditions (room temperature: 21 °C ± 2 °C, relative humidity: 50–60%, 12 h light/12 h dark cycle) and had free access to liquid diet LD 101 for rodents (PMI LabDiet, Richmond, VA, USA). Animal handling and experimental design procedures were approved by the First Local Ethics Committee for Animal Experiments in Wroclaw, Poland (approval No. 46/2017).

The rats were randomly divided into five equal groups: control (C), Cd, Cd + α-LA, Cd + Mg, and Cd + Mg + α-LA, and were treated accordingly (Table 1).

The selection of the experimental dose of cadmium was based on our previous studies [14], as well as on the reports of other authors [15,16]. Continuous exposure of rats to cadmium concentrations in the range of 5–50 mg Cd/L of drinking water reflects the level of human exposure to this metal; that is, from moderate to relatively high (e.g., cigarette smokers, professionals exposed to Cd) [15]. In our previous experiment [14], the dose of 20 mg Cd/L administered with a liquid diet for five months induced changes in bone morphology and metabolism, as well as disorders in hematological parameters and organ functioning (data not published). In the presented study, in order to induce similar changes, but within a shorter time (three months), we decided on a higher dose (30 mg Cd/L), but remaining within the scope of occupational and high environmental exposures. Literature data report that α-lipoic acid administered to rats at doses of 25–100 mg/kg of body weight increases GSH levels and protects kidneys from cisplatin-related damage [12]. Intraperitoneal administration of α-lipoic acid to older rats at a dose of 100 mg/kg body weight for 7 and 14 days increased the level of antioxidants (vitamin C, vitamin E, and reduced glutathione) and reduced the level of lipid peroxidation in the brain [13]. In order to reduce the animals’ stress, we administered α-lipoic acid orally, and the applied dose was comparable with the therapeutic doses used in the above-mentioned studies and equaled 100 mg/kg body weight, four times per week. Magnesium was added to liquid feed at a dose of 150 mg Mg/L, which corresponds to the content of this element in good mineral waters available on the market (recommended in health prevention) [17]. The recommended dose of Mg for adult men equals 400–420 mg daily (6 mg/kg body weight), however, relatively higher doses can be used for specific medical problems [18].

The experiment was conducted for three months. Throughout this period, the consumption of the liquid diet was monitored daily with the crude method of determining the volume of the fluids remaining in the bottles as a measure of the previous day’s intake. The animals were weighed every week. The mean consumption of liquid feed, Cd, Mg, and α-LA was calculated and converted to kg body mass per week. At the end of the study, the rats were anaesthetized by an intramuscular ketamine + xylazine injection (10 mg xylazine and 50 mg ketamine/kg of body weight). Blood samples were collected via tubes after a cardiac puncture, and then the animals were sacrificed through the dislocation of cervical spine. Separate portions of blood were transferred to tubes without anticoagulant and tubes containing either lithium heparin or trisodium EDTA (Trisodium Ethylenediamine Tetraacetate) as anticoagulants. Serum was prepared by centrifugation (3000× *g*, 10 min) from the samples of blood without anticoagulant and used for biochemistry analysis. EDTA blood was analyzed for hematology. Serum and heparinized samples of blood were used for bioelements and cadmium analysis, respectively. Kidney and liver were removed, washed with cold saline solution, cut into pieces, and stored at −70 °C until used. The femurs were collected, and the soft tissues such as muscle pieces were removed manually. Bones were stored in a freezer in plastic packaging.

### 2.3. Biochemical and Hematological Analysis

#### 2.3.1. Determination of Hematological Parameters

Blood parameters, including white blood cell (WBC) and red blood cell (RBC) counts, hemoglobin (HGB), hematocrit (HCT), mean corpuscular volume (MCV), mean corpuscular hemoglobin (MCH), mean corpuscular hemoglobin concentration (MCHC), red blood cell distribution width (RDW), and platelets (PLT), were determined with the use of an automated hematological analyzer Sysmex XN-2000™ (Sysmex Co. Ltd., Kobe, Japan).

#### 2.3.2. Serum Biochemical Analyses

##### Serum Lipids and Markers of Liver and Kidney Dysfunction

Biochemical markers of liver and kidney dysfunction and serum lipids were measured using an automatic biochemistry analyzer Architect ci4100 (Abbott Diagnostics, Lake Forest, IL, USA) and commercial tests of the same manufacturer. The measured parameters included aspartate aminotransferase (AST), alanine aminotransferase (ALT), γ-glutamyl transpeptidase (GGTP), lactate dehydrogenase (LDH), urea and calculated urea nitrogen (BUN), creatinine (CRE), and uric acid (UA). Serum lipids included total cholesterol (CHOL), high-density lipoprotein (HDL), and triglycerides.

##### Measurement of C-Reactive Protein, Bone Turnover Markers, and Vitamin D_3_

The levels of osteocalcin (OC, bone formation marker), C-terminal telopeptide of type I collagen (CTX-1, bone resorption marker), and 1,25-dihydroxyvitamin D_3_ (1,25(OH)2D_3_) in serum were measured with an enzyme-linked immunosorbent assay (ELISA) method using the SEA471Ra, CEA665Ra, and CEA467 Ge kits, respectively (USCN Life Science Company, Wuhan, Hubei, China). Serum C-reactive protein (CRP) was quantified by the Rat C-Reactive Protein ELISA kit Catalog No. 80670 (Crystal Chem Inc., Grove Village, IL, USA). The measurements were performed in accordance with the manufacturers’ recommendations with the use of a plate reader (PowerWave XS, BioTek Instruments, Winooski, VT, USA).

#### 2.3.3. Measurement of Oxidative Stress and Antioxidant Defense in Serum, Kidney, and Liver

##### Thiobarbituric Acid Reactive Substances (TBARs)

The concentration of thiobarbituric acid reactive substances (TBARs) was used as marker of oxidative stress in kidney, liver, and serum. Tissues were homogenized in a radioimmunoprecipitation (RIPA) buffer (Sigma Aldrich, Cat No. R-0278) in a ratio of 1:10 *w*/*v* and then centrifuged at 1600× *g* and 4 °C for 10 min. The measurement was performed on the serum and supernatant of tissues using the thiobarbituric acid reactive substances (TBARs) assay kit No. 10009055 (Cayman Chemical Company, Ann Arbor, MI, USA) in accordance with the manufacturer’s instructions.

##### Total Antioxidant Capacity 

Total antioxidant capacity was measured in the serum of rats with the Antioxidant Assay Kit No. 709001 (Cayman Chemical Company, Ann Arbor, MI, USA) in accordance with the manufacturer’s instructions. The assay relies on the ability of antioxidants in a sample to inhibit the oxidation of 2,2′-azino-di-(3-ethylbenzthiazoline sulphonate) (ABTS) by metmyoglobin. The capacity of the antioxidants in the serum to prevent ABTS oxidation was compared with that of Trolox (a water-soluble tocopherol analogue) and quantified as mM Trolox equivalents.

##### Reduced Glutathione

The concentration of reduced glutathione (GSH) in liver and kidney was determined using the Glutathione Assay Kit No. 703002 (Cayman Chemical Company, Ann Arbor, MI, USA). Tissues were homogenized in 5 mL cold buffer (50 mM MES, pH 6.0, containing 1 mmol/l EDTA) per g of tissue. Next, equal volumes of metaphosphoric acid (No. 239275, Sigma-Aldrich, Saint Louis, MO, USA) solution (5 g metaphosphoric acid/50 mL water) were added and homogenates were centrifuged at 2000× *g* for 2 min. Deproteinized supernatants were used for assay.

##### Antioxidant Enzyme Activity in Liver and Kidney

(a)Glutathione peroxidase (GPx) assay

Tissues were homogenized in a cold buffer (50 mM TRIS-HCl buffer, pH 7.5, containing 5 mM of EDTA and 1 mM of 2-mercaptoethanol) and centrifuged at 10,000× *g* at 4 °C for 10 min. GPx activity was measured in the tissues supernatants using the Bioxytech^®^ GPx-340 kit No. 21017 (Oxis International, Portland, OR, USA) in accordance with the manufacturer’s instructions. GPx activity was expressed as mU GPx per mg of protein (1 mU/mg = 1 nmol of oxidized NADPH in 1 min per mg of protein).

(b)Superoxide dismutase (SOD) assay

Tissues were homogenized in a cold, 20 mM HEPES (4-(2-hydroxyethyl)-1-piperazineethanesulfonic acid) buffer, pH 7.2, containing 1 mM EGTA (ethylene glycol-bis(2-aminoethylether)-N,N,N′,N′-tetraacetic acid), 210 mM mannitol, and 70 mM sucrose. The homogenates were centrifuged at 1500× *g* at 4 °C for 5 min. SOD activity was measured in the obtained supernatants with the Cayman Chemical Company kit No. 706002 in accordance with the manufacturer’s instructions. One unit of SOD was defined as the amount of enzyme decreasing superoxide anion concentration by 50%.

(c)Catalase (CAT) assay

Tissues for catalase analysis were homogenized in a cold 50 mM PBS (phosphate-buffered saline) buffer supplemented with 1 mM EDTA per g of tissue. The homogenates were centrifuged at 10,000× *g* at 4 °C for 15 min. The enzyme activity was measured in the supernatants using kit No. STA-341(Cell Biolabs, San Diego, CA, USA) in accordance with the manufacturer’s instructions. One unit of CAT was defined as the amount of enzyme decomposing 1 mM of H_2_O_2_ in 1 min at 25 °C.

The enzymes’ activities were expressed as activity units per mg of protein. Protein concentrations in the samples were measured with the Lowry’s method (24) using a kit from the Sigma Diagnostics company. Bovine albumin was applied as the standard.

Determinations of markers of oxidative stress and antioxidant potential were performed with the use of a plate reader (PowerWave XS, BioTek Instruments, Winooski, VT, USA).

#### 2.3.4. Determination of Cadmium Content in the Femurs and Blood

The concentration of Cd in the bones was determined by atomic absorption spectroscopy. The weighed mass (0.1–0.3 g) of the dried tissue was placed in a perfluoroalkoxy (PFA) vial with a screw cap. Then, 1.5 mL of concentrated nitric acid (Baker Analyzed Instra) was added to the samples and chemically decomposed with a heating block at 120 °C for 16 h. After cooling, 1.5 mL of 30% hydrogen peroxide (Baker Analyzed) was added, and the mineralization was continued for another 24 h. The resulting solution was diluted with deionized water (Milli-Q, Millipore) to 16 mL. A sample of reference material and blanks was prepared with the same procedure. Cd concentration was determined by flame atomic absorption spectroscopy (FAAS) in an airacetylene flame using an atomic absorption spectrometer (UNICAM 939 Solaar, Cambridge, UK). The results are expressed in µg per g dry weight.

Cadmium in the blood was determined by flameless atomic absorption spectrometry with atomization in a THGA (Transversely-Heated Graphite Atomizer) type graphite furnace and Zeeman background correction, using a Perkin-Elmer spectrometer 4100ZL (Bodenseewerk Perkin-Elmer, Ueberlingen, Germany). For the analysis, a modified Stoeppler and Brandt [19] method was used. Cadmium concentration was determined in a sample of blood deproteinized with nitric acid at a concentration of 0.8 M. The concentration of cadmium in the sample was obtained by comparing the absorbance of the sample with a standard curve made using standard solutions prepared in bovine blood. The detection limit of the described method was 0.16 µg/L.

#### 2.3.5. Determination of Serum Concentration of Calcium (Ca), Magnesium (Mg), Iron (Fe), and Zinc (Zn)

Ca, Mg, Fe, and Zn in the serum were determined by flame atomic absorption spectroscopy (FAAS) in an airacetylene flame using an atomic absorption spectrometer (UNICAM 939 Solaar, Cambridge, UK) equipped with deuterium background correction. Serum samples for the determination of Ca and Mg were diluted 50-fold with a 0.2% solution of lanthanum (LaCl_3_) as the eluting buffer in the presence of phosphates. For the determination of Zn and Fe, the serum was diluted 10-fold with deionized water. Calibration was carried out with aqueous standard solutions. The basic parameters for the measurement of the individual elements are shown in Table 2.

### 2.4. Statistical Analysis

The results were expressed as means (±SD). The distribution and variance of all data were examined using the Shapiro–Wilk and Levene’s test, respectively. Data with homogenous variance were analyzed with one-way analysis of variance (ANOVA), followed by near infrared (NIR) post-hoc test. The Cochran–Cox test served to determine essential differences between the groups of data with heterogeneous variances. The Mann–Whitney test was used for nonparametric data. Pearson’s and Spearman’s (for non-parametric data) correlation analyses were conducted to examine the relationship between the measured parameters. The data were analyzed statistically with the Statistica 13.1 for Windows software (StatSoft, Krakow, Poland). Statistical significance was set at *p* < 0.05.

## 3. Results

### 3.1. Body Weight Gain; Consumption of Liquid Diet; and Intake of Cd, Mg, and α-LA

Table 3 shows mean body weight gain, feed and α-lipoic acid consumption, and calculated Cd and Mg intake by the rats during the three-month experiment.

The average daily feed consumption was significantly lower (by about 12%) in the group exposed only to cadmium (Cd) than in the control (C) group and in groups supplemented with magnesium and/or α-lipoic acid (Cd + Mg, Cd + α-LA, and Cd + Mg + α-LA). In all groups exposed to cadmium (Cd, Cd + α-LA, Cd + Mg, and Cd + Mg + α-LA), the rats showed a significant suppression of body weight gain (decreased by 39%, 33%, 34%, and 26%, respectively) as compared with the control animals (*p* < 0.05). The daily dose of Cd in the Cd-exposed groups was 6.9–7.6 mg/kg body weight, whereas the mean intake of magnesium in rats supplemented with Mg equaled 38.0 (±1.98) mg/kg body weight. The dose of cadmium associated with the volume of feed consumed was the lowest in animals treated only with cadmium (Cd group).

### 3.2. Hematological Parameters

The effects of Cd treatment and supplementation with Mg and/or α-LA on hematological parameters are summarized in Table 4 and Appendix A.

Administration of cadmium to rats (Cd group) resulted in a significant increase (almost twofold) in the number of WBC compared with the control. In groups Cd + Mg and Cd + Mg + α-LA, the number of WBC was also significantly elevated, while there was no significant difference between the C group and the group supplemented only with α-lipoic acid (Cd + α-LA group).

The hemoglobin concentration (HGB), mean corpuscular hemoglobin content (MCH), mean corpuscular volume (MCV), hematocrit (HCT), and platelets counts (PLT) were significantly lower (by about 33%, 18%, 19%, 33%, and 27%, respectively), whereas red blood cell distribution width (RDW) was significantly higher (by 35%) in rats treated with cadmium (Cd group) compared with the control (*p* < 0.05). Supplementation with α-LA and/or Mg tended to increase the HGB level in comparison with the group treated with cadmium alone, but only in the case of Cd + α-LA group was the difference was statistically significant, and the HGB value was still substantially lower than that in the control group (*p* < 0.05). In the groups of rats that were supplemented with Mg or Mg + α-LA, the mean MCH and PLT were similar, whereas the mean corpuscular hemoglobin concentration (MCHC) was even higher than that in the C group.

### 3.3. Biochemical Markers of Liver and Kidney Dysfunction

Serum biochemical parameters (liver and kidney function biomarkers) in rats exposed to cadmium and supplemented with α-lipoic acid and/or Mg are presented in Table 5.

Administration of cadmium with liquid diet for three months resulted in a significant (*p* < 0.05) increase in the activities of aspartate aminotransferase (AST), alanine aminotransferase (ALT), γ-glutamyl transpeptidase (GGTP), and lactate dehydrogenase (LDH) in the rat serum. Supplementing with magnesium and especially Mg together with α-lipoic acid decreased LDH and AST activities to a value close to the control. GGTP activity decreased and normalized also as a result of Mg and/or α-LA administration.

A significant decline of the serum urea concentration and values of the BUN/creatinine ratio (by 24% and 19%, respectively) was observed in rats treated with cadmium in comparison with the control animals, *p* < 0.05. Administration of α-LA and Mg separately did not affect the normalization of urea concentration. However, the combined action of Mg and α-LA acid restored both urea concentration and the value of the BUN/creatinine ratio to control levels. The levels of the BUN/creatinine ratio in rats exposed to Cd and co-supplemented with Mg and α-LA were significantly higher than in those receiving only cadmium (Cd group), cadmium with Mg (Cd + Mg group), or cadmium with α-LA alone (Cd + α-LA group) (by about 28%, 26%, and 18%, respectively) (*p* < 0.05).

### 3.4. Effects on Serum Lipid Profile

Serum lipid parameters including total cholesterol (CHOL), high-density lipoprotein cholesterol (HDL), triglycerides (TRIG), and the CHOL/HDL index are shown in Table 6.

The concentration of CHOL in serum significantly decreased in the Cd, Cd + α-LA, and Cd + Mg groups when compared with the control value (by 16%, 28%, and 21%, respectively; *p* < 0.05). However, in the group supplemented with Mg and α-LA simultaneously, the serum concentration of CHOL was similar to the control. The greatest level of CHOL/HDL and the lowest concentration of HDL in comparison with other groups were also observed in rats co-supplemented with Mg and α-LA.

### 3.5. Serum C-Reactive Protein Level

The effect of cadmium treatment and supplementation with Mg and/or α-LA on the serum CRP level is shown in Figure 1.

CRP is a reliable marker of inflammation that raises in response to different inflammatory stimuli. Cd treatment had a significant effect on increasing CRP levels in the serum of rats (by 44%). Administration of Mg or combined Mg and α-LA significantly decreased the serum level of CRP when compared with treatment with cadmium alone (by 17% and 34%, respectively; *p* < 0.05). The serum C-reactive protein level was significantly lower in rats from the Cd + Mg + α-LA group when compared with the Cd + α-LA and Cd + Mg groups (by 21% and 10%, respectively; *p* < 0.05).

### 3.6. Serum Bone Turnover Markers and 1,25-dihydroxyvitamin D_3_ Levels

Figure 2 presents the level of osteocalcin (OC), C-terminal telopeptide of type I collagen (CTX-1), and 1,25-dihydroxyvitamin D_3_ in the rat serum.

In rats exposed to cadmium, the level of CTX-1 (bone resorption marker) was significantly higher (by 45%), whereas the levels of vitamin D_3_ and OC (bone formation marker) were lower (by 17% and 24%, respectively; *p* < 0.05) than those in the control group. Ingestion of the supplements (Mg and/or α-LA) significantly reduced the CTX-1 level and increased the serum levels of OC and vitamin D_3_ when compared with those in rats administrated with cadmium alone (*p* < 0.05) and normalized the values of these parameters to the control levels. The concentrations of OC in rats from the Cd + Mg + α-LA group were significantly higher than those in rats in all other experimental groups (*p* < 0.05).

### 3.7. Markers of Oxidative Stress and Antioxidant Potential

#### 3.7.1. Thiobarbituric Acid Reactive Substances (TBARs) Concentration and Antioxidant Potential in Serum

Figure 3 shows the effect of cadmium treatment and supplementation with Mg and/or α-LA on the level of oxidative stress (as expressed by TBARs concentration) and antioxidant potential in the serum of rats.

(a)The level of the lipid peroxidation marker (TBARs) was significantly higher in the serum of rats exposed to cadmium alone (Cd) compared with the C group (41.1 µM/L vs. 19.4 µM/L) and with the Cd + α-LA and Cd + α-LA + Mg groups (by 22% and 45%, respectively; *p* < 0.05).(b)We also noticed that the antioxidant potential of serum from cadmium rats (Cd group) dropped and equaled approximately 80% of that in control rats. However, this decline was reversed as a result of α-lipoic acid supplementation. The serum collected from the Cd + α-LA group showed a significant increase in antioxidant activity, by about 45%, when compared with the Cd group and group supplemented with magnesium alone (Cd + Mg), *p* < 0.05.

#### 3.7.2. TBARs, Reduced Glutathione, and Antioxidant Enzymes in Rats’ Liver and Kidney

Table 7 shows the effect of cadmium treatment and supplementation with Mg and/or α-LA on the level of oxidative stress (as expressed by TBARs concentration), reduced glutathione (GSH) concentration, and the activities of antioxidant enzymes in the rats’ liver and kidney.

(a)Thiobarbituric acid reactive substances (TBARs)

Similar to in the case of serum, the liver and kidney concentrations of TBARs were significantly higher in the rats from the cadmium-intoxicated group (Cd) compared with the control level (by 24% and 62%, respectively; *p* < 0.05). Supplementation of Mg and/or α-LA significantly reduced the concentration of TBARs in liver and kidney and restored the levels of lipid peroxidation in these organs to those of control animals. In the groups supplemented with a combination of Mg and α-lipoic acid, the kidney level of TBARs was significantly lower than that in groups supplemented with α-lipoic acid or Mg separately (by 21% and 8%, respectively; *p* < 0.05).

(b)Reduced glutathione (GSH) and antioxidant enzymes

GSH functions as a substrate for glutathione peroxidase and shows antioxidant properties itself. The superoxide dismutase (SOD) enzyme converts superoxide radicals into a less toxic product (H_2_O_2_) and is the first line in cell defense against oxidative stress. Catalase (CAT) is an antioxidant enzyme that counteracts the increase of hydrogen peroxide concentration, rapidly converting it into water and oxygen gas. The cellular glutathione peroxidase (GPx) is an enzyme whose biochemical function (similarly as in CAT) consists of reducing free hydrogen peroxide to water, as well as neutralizing lipid peroxides [20].

(c)GSH

The glutathione concentration decreased significantly in the liver and kidney of rats exposed to Cd alone when compared with the control level (by 20% and 30%, respectively; *p* < 0.05). Supplementing with Mg and/or α-LA increased the GSH concentration in liver to a value close to the control. The GSH content in the liver of rats from Cd group was significantly lower (by 26%; *p* < 0.05) than that in the group co-supplemented with Mg and + α-LA. In the kidney, the administration of Mg or α-LA separately caused a substantial increase of GSH concentration in comparison with the group exposed only to cadmium; however, these levels were still significantly lower than those found in kidneys from control rats. The concentrations of GSH in the kidneys of rats from groups co-supplemented with Mg and α-LA were significantly higher than those in animals from all other cadmium exposed groups (*p* < 0.05).

(d)SOD

The results showed that the liver and kidney activity of SOD was substantially increased in rats from the Cd group in comparison with control animals (by 87% and 52%, respectively; *p* < 0.05). Supplementation with α-lipoic acid (Cd + α-LA and Cd + Mg + α-LA groups) restored the liver activity of SOD to the level of control. In the livers from rats supplemented only with Mg (Cd + Mg group), the activity of SOD was higher in comparison with that in the C group (by 87%; *p* < 0.05) and in animals from Cd + α-LA and Cd + Mg + α-LA groups (by 59% and 39%, respectively; *p* < 0.05). In the kidney from rats supplemented separately with Mg or α-LA, the activity of SOD was statistically significantly decreased, whereas in animals fed with both supplements, it was similar when compared with the rats exposed only to Cd.

(e)CAT

The liver activity of CAT was significantly higher in the Cd, Cd + Mg, and Cd + Mg + α-LA groups compared with control (by about 34%, 30%, and 17%, respectively; *p* < 0.05). However, in the Cd + α-LA group, the activity of this enzyme was similar to that in control animals.

Kidney CAT activity did not differ significantly between the exposed animals (Cd, Cd + Mg, Cd + α-LA, and Cd + Mg + α-LA groups) and control.

(f)GPx

The liver activity of GPx was above twice as high in the Cd group compared with control. GPx activities were significantly lower in the cadmium rats supplemented with Mg or α-LA (Cd + Mg, Cd + α-LA groups) than in the rats exposed to cadmium alone (by 38% and 35%, respectively; *p* < 0.05); however, these were still significantly higher than those found in the liver of control rats. In contrast to the above, animals supplemented with α-lipoic acid and Mg simultaneously showed similar liver activity of GPx when compared with rats receiving the control diet. GPx activity in the Cd + Mg + α-LA group was significantly lower than in the Cd, Cd + α-LA, and Cd + Mg groups (by 160%, 70%, and 60%, respectively; *p* < 0.05). Unlike in the liver, the activity of GPx in the kidneys decreased during exposure to cadmium (by nearly 60%; *p* < 0.05). However, its activity was significantly higher in the Cd + Mg and Cd + α-LA + Mg groups compared with the Cd group (by 84% and 92%, respectively; *p* < 0.05).

### 3.8. Cadmium Content in Bone and in the Blood and Concentration of Elements (Cd, Mg, Zn, P, Fe, and Ca) in the Serum of Rats

#### 3.8.1. Cadmium Concentration in Blood and Bone

Figure 4 illustrates the femur and blood Cd concentrations by the groups of rats. In the blood and bones of rats, the Cd levels were significantly higher in the Cd-treated groups than in the control group. Cadmium concentrations in femurs did not differ significantly between the Cd-treated groups. The α-lipoic acid alone supplemented group (Cd + α-LA) showed slight, but statistically significant reductions in blood Cd levels in comparison with the cadmium alone treated group (Cd).

#### 3.8.2. Concentration of Elements (Mg, Zn, P, Fe, and Ca) in the Serum of Rats

As shown in Table 8, the concentration of iron (Fe) and zinc (Zn) in serum significantly decreased, and that of magnesium (Mg) and phosphorus (P) increased in the Cd group when compared with the control value (*p* < 0.05). Administration of Mg alone to rats exposed to Cd normalized the level of serum phosphorus only. Supplementing Mg + α-lipoic acid (Cd + Mg + α-LA group) normalized the Mg and P concentration in serum to a value close to the control, but the levels of Fe and Zn were still depressed. However, in the group supplemented only with α-lipoic acid (Cd + α-LA group), the serum levels of Fe and Zn were higher than those in the group treated with Cd alone (Cd group), and the levels of Zn, Mg, and P were similar to the control values.

### 3.9. Correlations between Cd in Blood and Bone, and Other Measured Parameters 

The blood cadmium concentration correlated positively with the activity of liver enzymes AST (*r* = 0.587, *p* < 0.01), ALT (*r* = 0.410, *p* < 0.05), and LDH (*r* = 0.513, *p* < 0.01), and negatively with total cholesterol concentration (*r* = −0.414, *p* < 0.05) and the CHOL/HDL index (*r* = −0.415, *p* < 0.05). We observed a positive correlation between bone cadmium level and AST (*r* = 0.364, *p* < 0.05) and ALT (*r* = 0.363, *p* < 0.05) activity, and a negative correlation with serum CHOL (*r* = −0.421, *p* < 0.05) and HDL concentrations (*r* = −0.466, *p* < 0.01). A negative relationship was also found between body weight gain and blood and bone Cd level (*r* = −0.635, *p* < 0.01 and *r* = −0.484, *p* < 0.01, respectively). In turn, body weight gain correlated positively with blood urea nitrogen (*r* = 0.601, *p* < 0.01), the BUN/CRE index (*r* = 0.528, *p* < 0.01), serum CHOL concentration (*r* = 0.537, *p* < 0.01), and the CHOL/HDL index (*r* = 0.471, *p* < 0.01), and negatively with liver enzymes activity AST (*r* = −0.686, *p* < 0.01), ALT (*r* = −0.421, *p* < 0.05), and LDH (*r* = −0.612, *p* < 0.01). Moreover, a relationship between increased feed intake and a decrease in GGTP (*r* = −0.403, *p* < 0.05) and LDH (*r* = −0.455, *p* < 0.05) was noted.

We did observe a positive correlation between blood and bone Cd concentration and WBC (*r* = 0.363, *p* < 0.05 and *r* = 0.362, *p* < 0.05, respectively), as well as anisocytosis (RDW-CV%) (*r* = 0.402, *p* < 0.05 and *r* = 0.682, *p* < 0.01, respectively); in turn, the correlation of bone Cd with red blood cell indices such as RBC, HGB, HCT, MCV, MCH, and MCHC was negative (*r* = −0.401, *p* < 0.05; *r* = −0.551, *p* < 0.05; *r* = −0.511, *p* < 0.01; *r* = −0.681, *p* < 0.01; *r* = −0.401, *p* < 0.01; and *r* = −0.401, *p* < 0.05, respectively). Blood Cd correlated negatively with HGB (*r* = −0.501, *p* < 0.01), HCT (*r* = −0.411, *p* < 0.05), MCV (*r* = −0.490, *p* < 0.01, and MCH (*r* = −0.393, *p* < 0.05). A negative relationship was found between body weight gain and RDW-CV% (*r* = −0.643, *p* < 0.01), and a positive relationship with HGB (*r* = 0.513, *p* < 0.01), RBC (*r* = 0.413, *p* < 0.05), platelet count (*r* = 0.430, *p* < 0.05), HCT (*r* = 0.496, *p* < 0.01), and MCV (*r* = 0.524, *p* < 0.01).

Blood Cd correlated positively with serum levels of CRP (*r* = 0.375, *p* < 0.05), CTX-1 (*r* = 0.392, *p* < 0.05), Mg (*r* = 0.450, *p* < 0.05), and liver GPx activity (*r* = 0.584, *p* < 0.01), and negatively with renal GSH level (*r* = −0.528, *p* < 0.01) and serum levels of Zn (*r* = −0.625, *p* < 0.05) and Fe (*r* = −0.619, *p* < 0.05). A relationship was observed between blood and bone Cd concentration and the increase of SOD activity (*r* = 0.473, *p* < 0.01 and *r* = 0.374, *p* < 0.05, respectively) and decrease of GPx activity (*r* = −0.423, *p* < 0.05 and *r* = −0.371, *p* < 0.05, respectively) in kidneys. A relationship between the increased level of bone cadmium and a decrease in Fe (*r* = −0.464, *p* < 0.05) and Zn (*r* = −0.364, *p* < 0.05) in serum and increased serum Mg (*r* = 0.386, *p* < 0.05) was also noted. The volume of feed consumed correlated negatively with TBARs levels in the liver (*r* = −0.622, *p* < 0.05), as well as SOD (*r* = −0.413, *p* < 0.05) and CAT activity in the liver (*r* = −0.413, *p* < 0.05); it correlated positively with the levels of Fe (*r* = 0.509, *p* < 0.01) and vitamin D3 (*r* = 0.464, *p* < 0.01) in the serum, GSH levels in the liver (*r* = 0.362, *p* < 0.05) and kidneys (*r* = 0.465, *p* < 0.01), and CAT (*r* = 0.450, *p* < 0.05) and GPx activity (*r* = 0.500, *p* < 0.01) in the kidneys. CRP level correlated negatively with body weight gain (*r* = −0.439, *p* < 0.05) and the volume of feed consumed (*r* = −0.515, *p* < 0.01). Moreover, body weight gain correlated negatively with CTX-1 level (*r* = −0.397, *p* < 0.05) and positively with Fe (*r* = 0.472, *p* < 0.01), Zn (*r* = 0.586, *p* < 0.05) and Ca (*r* = 0.409, *p* < 0.01) levels in the serum.

## 4. Discussion

### 4.1. Effects of Cadmium Poisoning

The presented research, performed on an animal model, revealed a significant cadmium-induced inhibition of body mass increase. The effect was associated, among others, with reduced appetite and feed intake in animals exposed to cadmium. Early symptoms of iron-deficiency anemia were implied by a decrease in blood hemoglobin concentration, corpuscular hemoglobin, hematocrit, and corpuscular volume (MCV); by anisocytosis (increased red blood cell distribution width—corpuscular volume index, RDW-CV%); and by lowered serum iron, zinc, and calcium concentrations. A decreased platelet count (PLT) and serum total cholesterol were also observed. Cadmium-induced unfavorable changes in blood morphology are well documented in the available literature [21,22,23].

In the cadmium-exposed groups, the concentration of cadmium in the bones, reflecting the cumulative metal dose (the intake during the experiment), equaled 1.62–1.80 µg/g of bone and was comparable to that observed in jaw bone samples taken from people with environmental exposure to heavy metals (min–max range: 0.10–1.84 µg/g, Nowa Ruda, Poland) [24] and to that found in small rodents *(Myodes glareolus*) from contaminated industrial sites in Slovakia (range: 1.08–5.88 μg/g of bone) [25]. However, our study was designed to reflect a long-term exposure, and we are aware that blood cadmium concentration represents a recent exposure measure. The blood cadmium concentration was considerably higher in our experiment (range: 198–244 μg/L of blood) than that reported in people in developed countries, even in those occupationally exposed to cadmium (e.g., USA: 1.32 ± 0.37 μg/L in workers of a plastics factory, 7.9 ± 2.0 μg/L in workers of the metal recovery plant in Colorado; Poland: 2.98 ± 3.29 μg/L in steelworkers from Legnica and Głogów) [26]. The high blood cadmium concentration found in our animals is the result of the limited duration of the experiment and the use of relatively high doses of cadmium in order to produce clear biological effects. Nevertheless, these concentrations were comparable to those in people in developing countries, such as Nigeria (range: 130–620 μg Cd/L of blood in occupational exposure population [27]; 125.6 ± 24.0 μg Cd/L of blood in gas station employees, car mechanics, and so on, aged 31–40 years [28]), in which environmental and occupational heavy metal poisoning is, besides poor hygiene and infectious diseases, among the most important public health problems [27].

Cadmium intoxication increased (nearly twofold) blood leukocyte concentration and serum CRP (marker of inflammation) concentration. A literature review performed by Olszowski et al. [29] revealed that, in most observations and reports, cadmium induced an increase in the levels of inflammation markers, such as CRP, interleukin (IL)-6, TNF-α, IL-1, and IL-8. However, the influence of Cd on inflammatory markers may depend on study parameters such as the subjects’ age, sex, or cadmium dose, with rather high cadmium loads being conducive to these processes. Pollack et al. [30], in their study performed in healthy women of childbearing age exposed to relatively low cadmium concentrations, observed no correlation between the blood cadmium level and CRP concentrations. In turn, in a Swedish cohort study, in population of middle-aged (45–64 years) men and women, the levels of cardiovascular risk markers, including CRP, were highest in subjects with blood cadmium concentrations within the fourth quartile (range: 0.50–5.1 μg/L, median: 0.99 μg/L) [31]. The authors also observed that hazard ratios for all cardiovascular incidents were higher in patients within the fourth B-Cd quartile. It was also noted that the least number of women was in quartile I. In addition to higher levels of CRP, quartile IV was characterized by higher average levels of glycated hemoglobin (HbA1c), a higher rate of smokers, and a higher prevalence of low education and low physical activity. For other evaluated cardiovascular risk factors, such as LDL, HDL, triglycerides, diabetes, and blood pressure, the differences among the B-Cd quartiles were small. The authors suggest that in terms of lowering the frequency of cardiovascular incidents, it is reasonable to take measures to reduce exposure to cadmium, even in populations with moderate exposure [31].

In our model, with the cadmium load at the level of high environmental or occupational exposure, a statistically significant positive correlation was observed between the CRP level and blood cadmium concentration (*r* = 0.375, *p* < 0.05). We did not find, though, a significant correlation between bone Cd concentration and the CRP level. These results may suggest that the level of inflammatory marker is mainly associated with current exposure to toxic cadmium compounds.

In the present experiment, cadmium significantly increased the levels of oxidative stress index (TBARs) in the rats’ serum, kidneys, and liver, but decreased the levels of reduced glutathione in kidneys and liver. The highest TBARs increase (nearly by 100%) was observed in kidneys, which additionally confirms that they are particularly sensitive to cadmium toxicity. The relationship between cadmium exposure and lipid peroxidation, increased generation of aggressive free radicals, weakened antioxidant defense, and oxidative damage to macromolecules and cell membranes is quite well recognized and has been observed by many researchers [4,5,6,32,33,34].

Cadmium-induced inflammatory processes and oxidative stress have a considerable significance in the pathophysiology of numerous other chronic diseases such as cardiovascular diseases [4], renal diseases [6,7], psoriasis [35], or neoplasms [36]. Oxidative stress plays a key role in cadmium-induced insulin resistance, which increases the risk of metabolic disturbances and diabetes [34]. Cadmium is also associated with diseases of the osteoarticular system, decreased bone strength [4,37], and disorders of reproductive functions [32].

### 4.2. Attenuating Effects of Magnesium and α-Lipoic Acid against Cd toxicity

In our experiment, we investigated the protective potential of α-lipoic acid (α-LA, a strong antioxidant and metal chelator) [12,13] and magnesium (Mg, an antioxidant, an element of broad biological importance) [10,11] in alleviating cadmium-induced inflammatory and pro-oxidative processes, changes in blood morphology, disorders of bone metabolism, and renal and liver function.

#### 4.2.1. Influence on Cadmium Body Burden, Body Mass Gain, and Hematological Parameters

The literature proves that both Mg and α-LA inhibit the absorption of Cd by the body, affect its distribution, and reduce its concentration in tissues [10,11,12,13]. In our experiment, this effect, as well as other protective impacts of these substances, were partly masked as a result of increased feed intake by the supplemented animals and, consequently, relatively higher cadmium exposure. Nevertheless, the Cd concentration in the blood of the supplemented rats showed a downward trend compared with the concentration in the group receiving cadmium alone, but the difference was statistically significant only in the Cd + α-LA group.

As mentioned earlier, cadmium inhibited body mass gain in the animals. The administration of α-lipoic acid and/or Mg influenced the animals’ appetite and increased the volume of feed intake. Despite increased feed intake, no significant effect on body mass gain was found; in all rats exposed to Cd, body mass increase remained significantly lower than in control animals. In the supplemented groups, however, hematological indicators slightly improved. In these animals, a tendency towards hemoglobin concentration increase was observed, although it was statistically significant only in the Cd + α-LA group, as compared with animals receiving cadmium solely. In this group, normalization of hematocrit values was also found, but the platelet count was lowered. Supplementation did not increase the mean corpuscular volume (MCV) reduced by Cd. In animals receiving magnesium or Mg + α-lipoic acid, the RBC count and erythrocyte volume were decreased, but the mean corpuscular hemoglobin (MCH) and PLT turned out to be close to the values in the control group. Moreover, normalization of cholesterol concentration was noted, but only in the rats that received both supplements. In this group (Cd + Mg + α-LA), the atherogenicity index (CHOL/HDL), associated with the risk of the development of atherosclerotic lesions, was also significantly higher (by more than 20%) than the control value, although not exceeding the value of 4 (assumed as the upper limit of the norm).

The influence of α-lipoic acid on blood morphology under conditions of exposure to heavy metals was the subject of earlier studies [38,39]. Ghosh et al. [38] obtained a more efficient impact of α-lipoic acid on the improvement of hematological parameters (changed by subchronic arsenic intoxication) than that observed in our study. These authors observed that oral administration of α-lipoic acid at the daily dose of 25 mg/kg body mass restores the arsenic-decreased RBC, PLT, and HGB concentrations to control values. They also revealed normalization of MCV and a significant reduction in the percentage of abnormal erythrocytes (echinocytes and spherocytes), ascribing this improvement, among others, to the α-lipoic acid increasing the antioxidant potential in the animals’ blood [39]. Similarly, the alleviation by α-lipoic acid of the effects of short-term, acute exposure to metals on the blood morphology of rats was obtained by Nikolić et al. [39]. Lead and cadmium caused a significant decrease in RBC, HGB, and hematocrit values, but in groups supplemented with α-lipoic acid, these changes were partially inhibited and the measured parameters turned out to be close to control values. The protective effect of α-lipoic acid is explained by the presence of SH groups, which show high affinity to metal ions and, upon binding them, neutralize their toxic impact. The weaker effect of α-LA on the improvement of blood parameters in our experiment is probably related to different conditions of the experiment, such as longer exposure time, different doses, and route of administration. It is also of considerable importance that, in the study by Nikolić et al. [39], the intraperitoneal administration of α-lipoic acid allowed a substantial and quick increase of its blood concentration. As revealed in the literature, Mg has a beneficial effect on blood morphology under various pathological and physiological conditions. Magnesium supplementation in rats exposed to carbon tetrachloride inhibited the toxicity of this solvent and prevented changes in blood parameters such as WBC, RBC, Hb, MCHC, and PLT [40]. Oral supplementation with Mg prevents dehydration of erythrocytes and improves the function of the blood cell membrane in patients with sickle cell anemia [41]. The intake of magnesium by athletes increases of hemoglobin and erythrocyte concentration and improves their sports efficiency and performance [42]. We observed that Mg increased the mean corpuscular hemoglobin (MCH) and normalized the platelet count, but it seems that, under our experimental conditions, it was less effective than α-lipoic acid. We also found no significant interaction between these supplements to improve hematological parameters.

#### 4.2.2. Influence on Level of Oxidative Stress, Inflammatory Markers, and Antioxidant Potential

A much clearer protective effect of the investigated substances was found in relation to the reduction of oxidative processes. Both in serum and in the studied tissues, the administration of Mg and/or α-LA significantly decreased the level of cadmium-induced oxidative stress; in kidneys and liver, it increased the reduced glutathione concentration. In serum, the highest enhancement of antioxidant potential was observed in the α-lipoic acid supplementation group. The decrease in TBARs concentration associated with combined administration of Mg + α-LA was more pronounced in kidneys than in liver, and in both tissues, simultaneous intake of these supplements caused a significant strengthening of the effect in relation to the increase in GSH concentration. This phenomenon can be explained by the fact that both magnesium and α-lipoic acid are essential for the synthesis of glutathione. Magnesium is required in the first stage of glutathione biosynthesis (catalyzed by cysteine ligase); which is considered to limit the formation of GSH [43]. In turn, the increase in the concentration of reduced glutathione under the influence of α-lipoic acid results not only from the reduction of its oxidized form, but also from the constant supply of cysteine (CYS) to the cells. It is an amino acid whose deficiency limits the rate of glutathione production. The α-lipoic acid penetrating into cells is immediately reduced to dihydrolipoic acid, which is also a powerful antioxidant and has the ability to regenerate oxidized ascorbate, glutathione, coenzyme Q, and vitamin E. The dihydrolipoic acid is then released into the extracellular environment, where it is reoxidized, reducing extracellular cystine to cysteine. A quick uptake of CYS into the cell, significant increase of its pool, and a rise in glutathione production occur [12]. Our results are in line with the report by Arivazhagan et al. [13], who observed that administration of α-LA to old rats prevented age-related loss of reduced glutathione and reduced lipid peroxidation in blood, liver, kidneys, and brain.

Exposure to cadmium caused an increase in superoxide dismutase (SOD) activity in kidneys and liver, indicating a rise in peroxide radical generation in these tissues. As mentioned earlier, SOD decomposes peroxide radical, transforming it into hydrogen peroxide, which in turn is decomposed by catalase (CAT) or glutathione peroxidase (GPx) into water and molecular oxygen. Effective neutralization of free radicals, therefore, requires close cooperation of the main antioxidant enzymes [20]. In the liver of cadmium-exposed animals, an increase in both CAT and GPx activity and a relatively small rise in TBARs levels were observed. These results indicate relatively effective protection of this organ against the pro-oxidative impact of cadmium. In kidneys, in turn, despite the significant increase in SOD activity, that of GPx and CAT remained unchanged. This line of defense proved to be malfunctional, which resulted in a significant increase of oxidative stress in kidneys. In the liver and kidneys of animals supplemented with α-LA alone, the activity of SOD and CAT was lower than in the Cd group and close to the control value.

A lack of substantial changes in the activity of antioxidant enzymes, especially SOD and CAT, indicates reduced generation of free oxygen radicals, as confirmed by normalization of the MDA level. It seems obvious that such results should be associated with the animal supplementation with α-LA, a powerful antioxidant, whose presence, even under cadmium exposure conditions, may inhibit adverse oxidative reactions and maintain the redox balance in cells. The above observations imply that, in this case, there was a direct neutralization of free radicals or prevention of their excessive generation by α-LA. In the group supplemented with magnesium only, significant alleviation of oxidative stress in the body organs was also observed; but here, another defense mechanism was present, namely a significant increase in CAT activity. The beneficial effect on catalase activity is probably the result of the stimulating influence of Mg on this enzyme gene expression. Liu et al. [44] reported the presence of a positive correlation between the magnesium content in the diet and liver tissue and, on the other hand, the increase of m-RNA expression for catalase and its raised activity in this organ. A decrease in TBARs levels was also observed in the group supplemented with Mg and α-LA; here, however, the effect of both supplements was noted and, similar to in the Cd + Mg group, the CAT activity in the liver was increased.

The supplementation of α-lipoic acid normalized the white blood cell count, and the intake of Mg, α-LA, or Mg + α-LA lowered CRP concentrations in rats exposed to cadmium to the levels close to the control. Combined administration of both supplements strengthened the anti-inflammatory effect. CRP concentration in the Cd + Mg + α-LA group was clearly reduced in comparison with animals receiving these supplements separately (the difference was statistically significant for the Cd + α-LA group).

The effect of magnesium on the inhibition of inflammatory processes is confirmed, among others, by the meta-analysis performed by Dibaba et al. [45], based on seven cross-sectional studies, which showed the occurrence of a negative correlation between the dietary intake of magnesium and the level of CRP. Moreover, these authors’ qualitative evaluation of five intervention studies suggests a potential beneficial effect of Mg supplementation on the decrease in serum concentration of this protein. They believe that the importance of magnesium supplementation in the prevention of chronic diseases results, at least in part, from the inhibition of inflammatory processes. Similar results from a meta-analysis of eight prospective studies, confirming the influence of Mg supplementation on the decrease of CRP level in serum, were obtained by Mazidi et al. [46]. It is maintained that the beneficial properties of magnesium are related, among others, to the regulation of calcium metabolism. With magnesium deficiency, the level of extracellular magnesium ions decreases, which in turn leads to the stimulation of macrophages and inflow of calcium ions to cells such as adipocytes, as well as neuronal and peritoneal cells. This results in N-methyl-D-aspartate receptors (NMDAR) stimulation; opening of non-selective channels to cations; and, in consequence, a significant increase in calcium ions concentration in neuronal cells. As an outcome of these changes, neurotransmitters and proinflammatory cytokines such as IL-6 are released into the bloodstream. IL-6 is a signaling molecule that increases the release of CRP from the liver, which helps to prolong the inflammatory response in the body [45].

The meta-analysis of 11 randomized controlled clinical trials conducted by Saboori et al. [47] showed that supplementation with α-lipoic acid significantly reduced the CRP level, but only in patients with considerably elevated concentrations of this inflammation marker. In turn, similar studies performed by Haghighatdoost and Hariri [48] reveal that mainly high doses of α-LA (≥600 mg per day) are effective in reducing the levels of inflammatory mediators such as IL-6, TNF-α, and CRP. It is assumed that the most reliable mechanism of the action of α-LA consists of the inhibition of nuclear factor kappa B (NF-kB) activation induced by TNF-α. NF-kB has a fundamental influence on the expression of genes that participate in cell apoptosis processes and increase the production of inflammatory cytokines, including IL-6 and cyclooxygenase 2 (COX-2). In normal conditions, NF-kB is bound with the inhibitor of kappa B (IkB) and remains inactive. IkB phosphorylation with IkB kinase (IKK) causes its proteosomal degradation and the activation of NF-kB. The α-lipoic acid may weaken inflammatory processes by modulating the mitogen-activated protein kinases (MAPK) pathway and inhibiting IkB phosphorylation. As a strong antioxidant, α-lipoic acid regenerates, among others, vitamin E, which in turn inhibits the activity of protein kinase C, which also causes IkB phosphorylation [47]. The unfavorable effect of cadmium on calcium metabolism, the pro-oxidative activity, and interference with signaling processes in cells are at least partly responsible for its toxicity. Exposure to Cd activates the redox-sensitive MAPK pathway kinases, which affects the growth of inflammatory mediators [49]. Combined Mg and α-LA intake by complementing their antioxidant capacities may, more effectively than their separate administration, restore the redox balance in a cell and reverse these adverse processes.

#### 4.2.3. Influence on Bone Metabolism

We observed that cadmium exposure disturbed bone metabolism; intensified resorption processes (increase in CTX-1 concentration); inhibited bone formation (reduction of osteocalcin concentration); and lowered the serum level of Fe, Ca, Zn, and 1,25(OH)2D_3_ (1,25-hydroxycholecalciferol, calcitriol). Blood cadmium concentration positively correlated with the resorption marker level, which confirms the relationship between cadmium load and bone mass loss.

Cadmium also disturbed Mg metabolism. It is reported that Cd increases (through the urinary tract) the elimination of Mg from the body and reduces its blood concentration [10]. In our experiment, quite unexpectedly, we observed an increased serum concentration of Mg in rats from the Cd group compared with the control values. The magnesium balance in the body is strictly controlled by the dynamic interaction of intestinal absorption, exchange with bone, and excretion through kidneys. Approximately 60% of magnesium is stored in bones, 30% of which is exchangeable and functions as a reservoir to stabilize the serum concentration. [50]. Therefore, the effect of Cd on serum Mg levels may also depend on the exposure conditions, the associated organism response, and the triggering of specific compensation mechanisms. According to Elin [51], the serum Mg concentration does not always properly reflect its status in the body, and values in reference ranges may be associated with the compensatory release of Mg from bones and mask long-lasting, hidden deficiency. The administration of magnesium under deficit conditions results in an increased retention of this bioelement in the body and a significant increase in its serum concentration [51]. This is what we observed in our experiment in rats exposed to cadmium and supplemented with magnesium. The disturbance of mineral metabolism and bone demineralization by cadmium was confirmed by an increase in serum phosphorus concentration and a decrease in iron and zinc levels.

The increased bone turnover, especially bone resorption and decreased vitamin D_3_ (1,25-dihydroxyvitamin D_3_) level, observed in our study is among the described mechanisms of Cd toxicity responsible for the development of osteoporosis. Cd inhibits the differentiation of mesenchymal cells to osteoblasts, favoring their differentiation into adipocytes. In turn, it stimulates osteoclasts, increasing bone porosity [16,52]. Even with low exposures, there was a correlation between cadmium load and an increase in urine levels of deoxypyridinoline (DPD) (bone resorption marker) and a decrease in bone mineral density (BMD). This relationship was particularly pronounced among postmenopausal women, that is, a group very sensitive to the development of osteoporosis [37]. The influence of Cd on bones may also proceed indirectly, through toxic effects on other organs involved in bone mineralization (liver, kidneys).

Cd causes structural and functional damage to proximal tubules, including their mitochondria, which are the only sites of 1,25(OH)2D_3_ biosynthesis, where mitochondrial hydroxylases convert 25(OH)D_3_ into the active form, 1,25(OH)2D_3_. The active form of vitamin D_3_ is essential for osteoblasts to produce osteocalcin, the most important, non-collagenous bone tissue protein. Lowering the level of 1,25-dihydroxyvitamin D_3_ reduces absorption of Ca from the intestines and impairs bone mineralization [16].

Penetrating into bones, cadmium changes their mineral composition, disturbs the metabolism of calcium and phosphorus, and joins the structure of hydroxyapatite crystals. This influences bone properties and contributes to a decrease of bone strength. Binding to bone proteins, cadmium changes their biochemistry and interferes with metabolic processes in bone tissue [16].

Supplementation with α-lipoic acid and/or Mg inhibited cadmium-induced bone resorption processes, stimulated bone formation, and restored the concentration of active vitamin D_3_ to control values. Additionally, administration of Mg, as well as of Mg and α-lipoic acid, stimulated bone formation to a greater extent than supplementation with α-lipoic acid only, as implied by a significant increase in osteocalcin concentration.

The protective effect of Mg on bones, as well as under conditions of exposure to Cd, can be explained, among others, by the influence of Mg on vitamin D_3_ synthesis. All three hydroxylases (25-hydroxylase in the liver, and 1α-hydroxylase and 24-hydroxylase in kidneys) involved in the transformation of vitamin D_3_ precursors into its active form are regulated by magnesium ions. Also, the vitamin D_3_ binding protein is magnesium-dependent. An adequate magnesium supply reduces the risk of vitamin D_3_ deficiency [53]. In magnesium deficit conditions, Matsuzaki et al. [54] observed changes in the expression of m-RNA for hydroxylases metabolizing vitamin D_3_ and a decrease in m-RNA for type II Na/Pi cotransporters, regulating calcium and phosphorus metabolism. In turn, Rude et al. [55] indicate that magnesium exerts an immunomodulating effect on bone formation stimulation. In rat studies, the authors observed that very deep Mg dietary deficiency caused a growth in P substance, a neuropeptide stimulating the production of cytokines such as interleukin (IL)1β or TNF-α. An increased production of these cytokines resulted in a raised concentration of receptor activator of nuclear factor kB ligand (RANKL), which stimulates osteoclasts maturation and activity, and a lowered concentration of osteoprotegerin (OPG), which inhibits the process by binding with RANKL; consequently, bone resorption was intensified. In turn, the weakening of osteoclastogenesis by α-lipoic acid may result from the inhibition of NF-kB activation. As NF-kB can be induced by the growth of free radicals, suppressing oxidative stress and restoring redox balance can be crucial for the protective mechanism of α-LA action on bones [56].

#### 4.2.4. Influence on Markers of Liver and Kidney Function

The cadmium-induced functional impairment of liver and kidneys (responsible for mineral balance) was implied by an increase in the serum activity of transaminases (AST, ALT), lactate dehydrogenase (LDH), and gamma-glutamyltranspeptydase (GGTP). Administration of Mg and/or α-lipoic acid significantly decreased the activity of the mentioned enzymes, indicating the protective effect of these supplements, especially against cadmium hepatotoxicity. In the case of lactate dehydrogenase, administration of Mg and, to a greater extent, Mg combined with α-lipoic acid decreased the activity of this enzyme to values similar to control ones. The obtained results confirm the cooperation of Mg and α-lipoic acid in counteracting organ damage caused by cadmium.

Cadmium exposure led to a decreased serum urea concentration, as well as the urea/creatinine index and BUN/creatinine index. Cadmium-induced impairment of protein metabolism in the liver (decrease of protein catabolism) results, among others, in decreased urea synthesis. Separate administration of α-lipoic acid and Mg did not result in normalization of serum urea concentration, the urea/creatinine index, or the BUN/creatinine index. However, in their combined action, a significant improvement and a clear effect of additive synergy were observed, as indicated by a statistically significant increase in the values of these parameters and their restoration to control levels.

## 5. Conclusions

Despite the relatively higher doses of cadmium consumed with feed by animals supplemented with magnesium and/or α-lipoic acid, the toxic effects measured in this study were weakened compared with the changes observed in rats receiving only cadmium in feed. The obtained results imply that both magnesium and α-lipoic acid reduce the level of inflammatory marker; enhance the antioxidant potential; and protect tissues, especially the liver, against cadmium toxicity. However, the possible higher efficacy of the combined administration of both supplements depended on the nature of the toxic changes and the parameter under investigation. Supplementing α-lipoic acid alone turned out more efficient in counteracting the toxic effects of Cd on red blood cell indicators and iron metabolism than the combined administration of both supplements. The beneficial effect of combined Mg and α-lipoic acid administration was observed in the case of their impact on the decrease in CRP level, the activation of antioxidant defense, and the reduction of lipid peroxidation, as well as increasing weight gain and normalization of cholesterol and urea concentrations and the BUN/creatinine index; this implies their synergistic advantageous effect on protein metabolism and liver function. Similarly, the combined administration of Mg and α-lipoic acid resulted in a significant reduction of serum lactate dehydrogenase activity, which suggests their synergistic protective effect against organ damage induced by cadmium. The obtained results demonstrate that, by mutually supplementing the mechanisms of action, especially the antioxidative ones, the combined intake of magnesium and α-lipoic acid under the conditions of exposure to cadmium may increase the defense against its toxicity.

## Figures and Tables

**Figure 1 ijerph-16-04483-f001:**
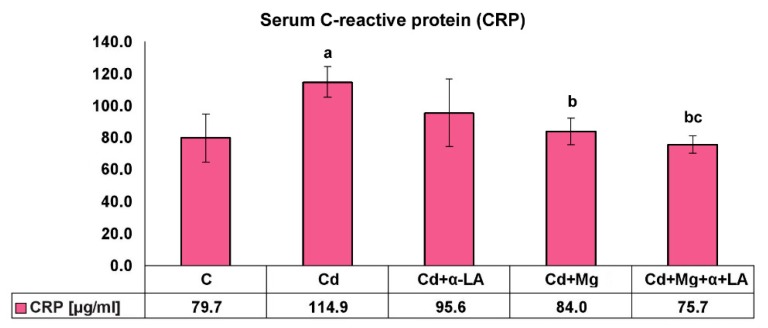
Effect of α-lipoic acid and/or magnesium supplementation on serum C-reactive protein (CRP) levels in cadmium-treated rats. Values are expressed as mean (±SD). ^a^ Significant change from control (C), *p* < 0.05; ^b^ group exposed to Cd separately significantly different from groups exposed to Cd and supplemented with Mg and/or α-lipoic acid (Cd vs. Cd + α-LA or Cd vs. Cd + Mg or Cd vs. Cd + Mg + α-LA), *p* < 0.05; ^c^ cadmium group supplemented with α-lipoic acid significantly different from cadmium group supplemented with Mg and cadmium group co-supplemented with Mg and α-lipoic acid (Cd + α-LA vs. Cd + Mg or Cd + α-LA vs. Cd + Mg + α-LA), *p* < 0.05.

**Figure 2 ijerph-16-04483-f002:**
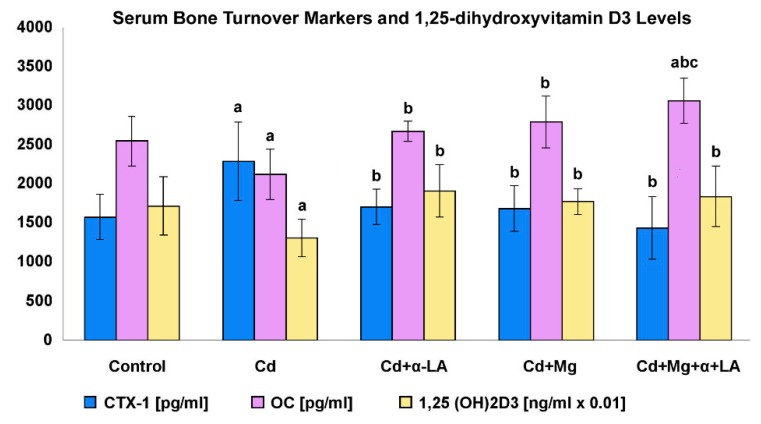
Effect of α-lipoic acid and/or magnesium supplementation on levels of bone turnover markers and vitamin D_3_ in the serum of rats treated with cadmium. Values are expressed as mean (±SD). Serum concentration of vitamin D_3_ are expressed as mean × 10^−2^ (±SD). OC, osteocalcin; CTX-1, C-terminal telopeptide of type I collagen; 1,25(OH)2D_3_), 1,25-dihydroxyvitamin D_3_; ^a^ significant change from control (C), *p* < 0.05; ^b^ group exposed to Cd separately significantly different from groups exposed to Cd and supplemented with Mg and/or α-lipoic (Cd vs. Cd + α-LA or Cd vs. Cd + Mg or Cd vs. Cd + Mg + α-LA), *p* < 0.05; ^c^ cadmium group supplemented with α-lipoic acid significantly different from cadmium group supplemented with Mg and cadmium group co-supplemented with Mg and α-lipoic acid (Cd + α-LA vs. Cd + Mg or Cd + α-LA vs. Cd + Mg + α-LA), *p* < 0.05.

**Figure 3 ijerph-16-04483-f003:**
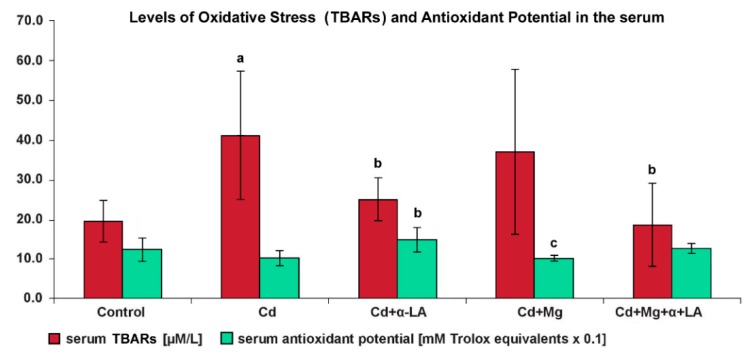
Effect of α-lipoic acid and/or magnesium supplementation on serum oxidative stress (thiobarbituric acid reactive substances, TBARs) and antioxidant potential levels in cadmium-treated rats. Values are expressed as mean (±SD). The serum level of antioxidant potential is expressed as mean × 10^−1^ (±SD). ^a^ Significant change from control (C), *p* < 0.05; ^b^ group exposed to Cd separately significantly different from groups exposed to Cd and supplemented with Mg and/or α-lipoic (Cd vs. Cd + α-LA or Cd vs. Cd + Mg or Cd vs. Cd + Mg + α-LA), *p* < 0.05. ^c^ cadmium group supplemented with α-lipoic acid significantly different from cadmium group supplemented with Mg and cadmium group co-supplemented with Mg and α-lipoic acid (Cd + α-LA vs. Cd + Mg or Cd + α-LA vs. Cd + Mg + α-LA), *p* < 0.05.

**Figure 4 ijerph-16-04483-f004:**
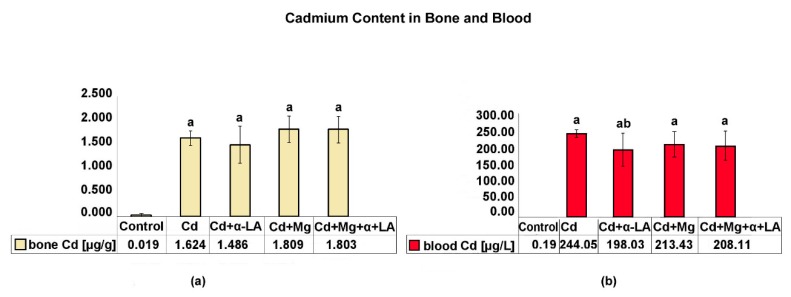
Effects of Cd treatment and supplementation with Mg and/or α-LA on the cadmium concentration: (**a**) in the bone (femur) and (**b**) in the blood. Values are expressed as mean (±SD); ^a^ significant change from control (C), *p* < 0.05; ^b^ group exposed to Cd separately significantly different from groups exposed to Cd and supplemented with Mg and/or α-lipoic (Cd vs. Cd + α-LA or Cd vs. Cd + Mg or Cd vs. Cd + Mg + α-LA), *p* < 0.05.

**Table 1 ijerph-16-04483-t001:** The protocol of the rats’ treatment.

Group of Rats	Treatment
C (control)	Rats were fed with liquid diet (LD101) (without any additives) ^1^
Cd (exposed to cadmium)	Rats were fed with liquid diet (LD 101) with the addition of cadmium dosed 30 mg Cd/kg (as CdCl_2_ × 2.5H_2_0) of feed ^2^
Cd + α-LA (exposed to cadmium and supplemented with α-lipoic acid)	Rats were fed with liquid diet (LD 101) with the addition of cadmium dosed 30 mg Cd (as CdCl_2_ × 2.5H_2_0)/kg of feed ^2^, and were supplemented with α-lipoic acid (100 mg/kg body weight, four times a week) ^3^
Cd + Mg (exposed to cadmium and supplemented with magnesium)	Rats were fed with liquid diet (LD 101) with the addition of cadmium dosed 30 mg Cd (as CdCl_2_ × 2.5H_2_0)/kg of feed and magnesium dosed 150 mg Mg (as C_3_H_7_MgO_6_P × H_2_O)/kg of feed ^4^
Cd + Mg + α-LA (exposed to cadmium and supplemented with magnesium and α-lipoic acid)	Rats were fed with liquid diet (LD 101) with the addition of cadmium dosed 30 mg Cd (as CdCl_2_x2.5H_2_0)/kg of feed and magnesium dosed 150 mg Mg (as C_3_H_7_MgO_6_P × H_2_O)/kg of feed ^4^, and were supplemented with α-lipoic acid (100 mg/kg body weight, four times a week) ^3^

Group of rats: C—control; Cd—exposed to cadmium; Cd + α-LA—exposed to cadmium and supplemented with α-lipoic acid; Cd + Mg—exposed to cadmium and supplemented with magnesium; Cd + Mg + α-LA—exposed to cadmium and supplemented with magnesium and α-lipoic acid). ^1^ Liquid diet for rodents (LD101) was prepared from the powder according to the manufacturer’s instructions. Fresh diet was prepared every two days and kept in a refrigerator; ^2^ during the preparation of the diet, a known amount of cadmium (30 mg Cd/kg diet) was added; ^3^ α-lipoic acid mixed with feed was given orally to rats in the amount of 0.4 mL using a syringe; ^4^ during the preparation of the diet, cadmium (30 mg Cd/kg diet) and magnesium (150 mg Mg/kg diet) were added.

**Table 2 ijerph-16-04483-t002:** The basic parameters for the measurement of Ca, Mg, Fe, and Zn.

Element	Wavelength (nm)	Gap (mm)	Background Correction	Flame Type
Ca	422.7	0.5	off	stoichiometric
Mg	285.2	0.5	on	stoichiometric
Zn	213.9	0.2	on	oxidizing
Fe	248.3	0.2	on	oxidizing

Ca—calcium; Mg—magnesium; Zn—zinc; Fe—iron.

**Table 3 ijerph-16-04483-t003:** Mean body weight gain and the consumption of liquid diet, α-lipoic acid (α-LA), cadmium, and magnesium by the groups of rats. ^1^.

Group of Rats	Body Weight Gain (g/rat) ^2^	Consumption of Liquid Diet (mL) ^3^	Cadmium Intake (mg) ^4^	α-Lipoic Acid Intake (mg) ^4,5^	Magnesium Intake (mg) ^4^
Control	226.2 (±31.4)	214.3 (±10.99)	–	–	–
Cd	137.3 (±24.8) ^a^	189.4 (±9.03) ^a^	6.87 (±0.33)	–	–
Cd + α-LA	149.3 (±28.6) ^a^	210.0 (±7.06) ^b^	7.63 (±0.26) ^b^	100	–
Cd + Mg	151.2 (±30.7) ^a^	210.0 (±12.95) ^b^	7.63 (±0.51) ^b^	–	38.1 (±2.5)
Cd + Mg + α-LA	166.3 (±10.5) ^a^	208.0 (±7.98) ^b^	7.58 (±0.3) ^b^	100	37.9 (±1.5)

Group of rats: C—control; Cd—exposed to cadmium; Cd + α-LA—exposed to cadmium and supplemented with α-lipoic acid; Cd + Mg—exposed to cadmium and supplemented with magnesium; Cd + Mg + α-LA—exposed to cadmium and supplemented with magnesium and α-lipoic acid). ^1^ Values are expressed as mean (±SD); ^2^ g per rat during the three-month experiment; ^3^ mL/kg body weight/24 h; ^4^ mg/kg body weight/24 h; ^5^ α-lipoic acid was given to rats only four times a week; ^a^ significant change from control (C), *p* < 0.05; ^b^ group exposed to Cd separately significantly different from groups exposed to Cd and supplemented with Mg and/or α-lipoic (Cd vs. Cd + α-LA or Cd vs. Cd + Mg or Cd vs. Cd + Mg + α-LA), *p* < 0.05.

**Table 4 ijerph-16-04483-t004:** Effect of cadmium treatment and supplementation with Mg and/or α-LA on the hematological profile in rats. ^1^.

	Control	Cd	Cd + α-LA	Cd + Mg	Cd + Mg + α-LA
WBC (count × 10^3^/μL)	4.97 (±1.05)	9.45 (±1.57) ^a^	5.28 (±1.07) ^b^	8.07 (±3.15) ^ac^	8.18 (±2.39) ^ac^
HGB (g/dL)	15.8 (±1.2)	10.52 (±2.7) ^a^	13.3 (±0.8) ^ab^	11.5 (±1.2) ^a^	11.6 (±1.1) ^a^
RBC (count × 10^3^/μL)	9.2 (±0.7)	7.5 (±2.0)	8.7 (±0.5)	6.2 (±2.2) ^ac^	7.1 (±2.3) ^a^
HCT (%)	48.3 (±4.5)	32.5 (±10.1) ^a^	38.8 (±4.0)	27.2 (±10.2) ^a^	30.4 (±9.5) ^a^
MCV (fl)	52.7 (±1.8)	42.5 (±3.2) ^a^	44.4 (±2.2) ^a^	43.8 (±1.5)	42.8 (±0.9) ^a^
MCH (pg)	17.2 (±0.50)	14.1 (±0.27) ^a^	15.2 (±0.14) ^a^	20.5 (±5.67) ^bc^	17.9 (±5.72)
MCHC (g/dL)	32.7 (±0.89)	33.4 (±3.20)	34.4 (±1.27) ^a^	46.9 (±13.66) ^abc^	41.6 (±1.8) ^abc^
RDW (fl)	18.8 (±1.19)	25.3 (±1.41) ^a^	25.5 (±2.12) ^a^	24.5 (±1.00) ^a^	25.5 (±0.88) ^abc^
PLT (count × 10^3^/μL)	870.3 (±78.2)	652.5 (±268.5) ^a^	582.7 (±207.2) ^a^	822.6 (±132.3) ^bc^	888.5 (±134.8) ^bc^

Group of rats: C—control; Cd—exposed to cadmium; Cd + α-LA—exposed to cadmium and supplemented with α-lipoic acid; Cd + Mg—exposed to cadmium and supplemented with magnesium; Cd + Mg + α-LA—exposed to cadmium and supplemented with magnesium and α-lipoic acid). ^1^ Values are expressed as mean (±SD); ^a^ significant change from control (C), *p* < 0.05; ^b^ group exposed to Cd separately significantly different from groups exposed to Cd and supplemented with Mg and/or α-lipoic (Cd vs. Cd + α-LA or Cd vs. Cd + Mg or Cd vs. Cd + Mg + α-LA), *p* < 0.05; ^c^ cadmium group supplemented with α-lipoic acid significantly different from cadmium group supplemented with Mg and cadmium group co-supplemented with Mg and α-lipoic acid (Cd + α-LA vs. Cd + Mg or Cd + α-LA vs. Cd + Mg + α-LA), *p* < 0.05. WBC—white blood cell; HGB—hemoglobin; RBC—red blood cell; HCT—hematocrit; MCV—mean corpuscular volume; MCH—mean corpuscular hemoglobin; MCHC—mean corpuscular hemoglobin concentration; RDW—red blood cell distribution width; PLT—platelets.

**Table 5 ijerph-16-04483-t005:** Effect of supplementation with Mg and/or α-LA on serum hepato-renal biomarkers. ^1^.

	Control	Cd	Cd + α-LA	Cd + Mg	Cd + Mg + α-LA
AST (U/L)	73.5 (±5.3)	356.8 (±357.3) ^a^	285.0 (±245.6)	155.3 (±77.6)	124.8 (±12.9)
ALT (U/L)	28.8 (±3.1)	137.8 (±85.3) ^a^	122.5 (±26.5) ^a^	96.2 (±16.7) ^a^	116.3 (±23.6) ^a^
GGTP (U/L)	4.0 (±0)	5.2 (±1.8) ^a^	4.0 (±0) ^b^	4.0 (±0) ^b^	4.0 (±0) ^b^
LDH (U/L)	290.3 (±114.2)	2404.8 (±1281.9) ^a^	1431.7 (±1132.8)	594.7 (±253.7) ^b^	400.7 (±155.6) ^bc^
CRE (mg/dL)	0.362 (±0.157)	0.327 (±0.039)	0.292 (±0.047)	0.308 (±0.024)	0.293 (±0.031)
Urea (mg/dL)	34.0 (±11.0)	25.7 (±3.7) ^a^	22.5 (±0.8) ^a^	26.2 (±2.8) ^a^	29.3 (±2.7)
BUN (mg/dL)]	15.9 (±5.1)	12.0 (±1.7) ^a^	10.5 (±0.4) ^a^	12.2 (±1.3) ^a^	13.7 (±1.3) ^c^
BUN/CRE	45.3 (±6.6)	36.7 (±2.8) ^a^	37.2 (±8.4) ^a^	39.8 (±4.1)	47.0 (±5.3) ^bcd^
Uric acid (mg/dL)	2.73 (±0.76)	3.32 (±2.95)	2.77 (±1.30)	3.13 (±0.87)	2.88 (±0.83)

Group of rats: C—control; Cd—exposed to cadmium; Cd + α-LA—exposed to cadmium and supplemented with α-lipoic acid; Cd + Mg—exposed to cadmium and supplemented with magnesium; Cd + Mg + α-LA—exposed to cadmium and supplemented with magnesium and α-lipoic acid; ^1^ Values are expressed as mean (±SD); ^a^ significant change from control (C), *p* < 0.05; ^b^ group exposed to Cd separately significantly different from groups exposed to Cd and supplemented with Mg and/or α-lipoic (Cd vs. Cd + α-LA or Cd vs. Cd + Mg or Cd vs. Cd + Mg + α-LA), *p* < 0.05; ^c^ cadmium group supplemented with α-lipoic acid significantly different from cadmium group supplemented with Mg and cadmium group co-supplemented with Mg and α-lipoic acid (Cd + α-LA vs. Cd + Mg or Cd + α-LA vs. Cd + Mg + α-LA), *p* < 0.05; ^d^ cadmium group supplemented only with Mg significantly different from cadmium group co-supplemented with Mg and α-lipoic acid (Cd + Mg vs. Cd + Mg + α-LA), *p* < 0.05. AST—aspartate aminotransferase; ALT—alanine aminotransferase; GGTP—γ-glutamyl transpeptidase; LDH—lactate dehydrogenase; CRE—creatinine; BUN—calculated urea nitrogen; BUN/CRE—calculated urea nitrogen/creatinine ratio.

**Table 6 ijerph-16-04483-t006:** Effect of Cd treatment and supplementation with Mg and/or α-LA on serum lipid profiles. ^1^.

	Control	Cd	Cd + α-LA	Cd + Mg	Cd + Mg + α-LA
CHOL (mg/dL)	67.5 (±10.8)	56.5 (±10.7) ^a^	48.5 (±8.0) ^a^	53.2 (±6.0) ^a^	58.2 (±7.7)
HDL (mg/dL)	21.5 (±3.9)	20.7 (±3.0)	18.7 (±3.1)	18.7 (±3.3)	15.7 (±3.1) ^ab^
TRIG (mg/dL)	120.2 (±45.6)	148.5 (±75.1)	101.5 (±21.2)	113.2 (±26.4)	137.3 (±66.5)
CHOL/HDL	3.16 (±0.19)	2.73 (±0.27)	2.61 (±0.28)	2.89 (±0.39)	3.87 (±1.16) ^abcd^

Group of rats: C—control; Cd—exposed to cadmium; Cd + α-LA—exposed to cadmium and supplemented with α-lipoic acid; Cd + Mg—exposed to cadmium and supplemented with magnesium; Cd + Mg + α-LA—exposed to cadmium and supplemented with magnesium and α-lipoic acid). ^1^ Values are expressed as mean (±SD); ^a^ significant change from control (C), *p* < 0.05; ^b^ group exposed to Cd separately significantly different from groups exposed to Cd and supplemented with Mg and/or α-lipoic (Cd vs. Cd + α-LA or Cd vs. Cd + Mg or Cd vs. Cd + Mg + α-LA), *p* < 0.05; ^c^ cadmium group supplemented with α-lipoic acid significantly different from cadmium group supplemented with Mg and cadmium group co-supplemented with Mg and α-lipoic acid (Cd + α-LA vs. Cd + Mg or Cd + α-LA vs. Cd + Mg + α-LA), *p* < 0.05; ^d^ cadmium group supplemented only with Mg significantly different from cadmium group co-supplemented with Mg and α-lipoic acid (Cd + Mg vs. Cd + Mg + α-LA), *p* < 0.05. CHOL—total cholesterol; HDL—high-density lipoprotein cholesterol; TRIG—triglycerides; CHOL/HDL—total cholesterol/ high-density lipoprotein cholesterol index

**Table 7 ijerph-16-04483-t007:** Effect of Cd treatment and supplementation with Mg and/or α-LA on the concentration of thiobarbituric acid reactive substances (TBARs) and reduced glutathione (GSH), as well as activities of antioxidant enzymes in rats’ liver and kidney. ^1^.

	Control	Cd	Cd + α-LA	Cd + Mg	Cd + Mg + α-LA
TBARs (ng/g tissue)					
Liver	423.3 (±96.6)	526.7 (±83.8) ^a^	415.0 (±48.2) ^b^	425.0 (±85.3) ^b^	431.7 (±50.9) ^b^
Kidney	273.4 (±57.3)	442.6 (±289.3) ^a^	295.9 (±24.8)	254.4 (±26.3) ^b^	233.6 (±61.2) ^bc^
GSH (mM/g tissue)					
Liver	19.0 (±2.3)	15.1 (±1.6) ^a^	17.2 (±3.1)	18.2 (±5.3)	20.3 (±4.8) ^b^
Kidney	2.61 (±0.43)	1.82 (±0.17) ^a^	2.13 (±0.15) ^ab^	2.13 (±0.10) ^ab^	2.77 (±0.81) ^bcd^
SOD (U_SOD_/mg protein)					
Liver	7.50 (±2.46)	14.01 (±3.06) ^a^	8.83 (±2.06) ^b^	14.03 (±3.95) ^ac^	10.29 (±02.33) ^bd^
Kidney	2.55 (±0.54)	3.89 (±0.60) ^a^	3.08 (±0.65) ^b^	3.21 (±0.57) ^b^	3.33 (±0.37) ^a^
CAT (U_CAT_/mg protein)					
Liver	1524.5 (±260.7)	2050.4 (±194.3) ^a^	1521.7 (±164.5) ^b^	1993.7 (±298.5) ^ab^	1790.7 (±135.8) ^ab^
Kidney	492.4 (±82.2)	482.4 (±67.7)	535.9 (±73.7)	510.4 (±53.7)	498.6 (±47.0)
GPx (mU_GPx_/mg protein)					
Liver	129.1 (±27.3)	334.5 (±75.3) ^a^	218.9 (±37.0) ^ab^	206.7 (±24.8) ^ab^	129.1 (±52.0) ^bcd^
Kidney	503.1 (±114.3)	214.1 (±118.5) ^a^	370.6 (±130.8)	394.6 (±148.0) ^b^	412.9 (±146.1) ^b^

Group of rats: C—control; Cd—exposed to cadmium; Cd + α-LA—exposed to cadmium and supplemented with α-lipoic acid; Cd + Mg—exposed to cadmium and supplemented with magnesium; Cd + Mg + α-LA—exposed to cadmium and supplemented with magnesium and α-lipoic acid). ^1^ Values are expressed as mean (±SD); ^a^ significant change from control (C), *p* < 0.05; ^b^ group exposed to Cd separately significantly different from groups exposed to Cd and supplemented with Mg and/or α-lipoic (Cd vs. Cd + α-LA or Cd vs. Cd + Mg or Cd vs. Cd + Mg + α-LA), *p* < 0.05; ^c^ cadmium group supplemented with α-lipoic acid significantly different from cadmium group supplemented with Mg and cadmium group co-supplemented with Mg and α-lipoic acid (Cd + α-LA vs. Cd + Mg or Cd + α-LA vs. Cd + Mg + α-LA), *p* < 0.05; ^d^ cadmium group supplemented only with Mg significantly different from cadmium group co-supplemented with Mg and α-lipoic acid (Cd + Mg vs. Cd + Mg + α-LA), *p* < 0.05. TBARs—thiobarbituric acid reactive substances; GSH—reduced glutathione; SOD—superoxide dismutase; CAT—catalase; GPx—glutathione peroxidase.

**Table 8 ijerph-16-04483-t008:** Effect of cadmium exposure and supplementation with Mg or/and α-LA on the serum concentration of iron, zinc, magnesium, phosphorus, and calcium ^1^.

	Control	Cd	Cd + α-LA	Cd + Mg	Cd + Mg + α-LA
Iron (mg/L)	2.42 (±0.33)	1.39 (±0.17) ^a^	1.92 (±0.36) ^ab^	1.77 (±0.71) ^a^	1.36 (±0.40) ^a^
Zinc (mg/L)	1.45 (±0.12)	1.01 (±0.10) ^a^	1.35 (±0.17) ^b^	1.00 (±0.06) ^a^	1.15 (±0.12) ^a^
Magnesium (mg/L)	26.2 (±3.85)	31.8 (±4.60) ^a^	30.6 (±5.90)	32.3 (±1.85) ^a^	30.9 (±4.55)
Phosphorus (mg/dL)	6.91 (±1.08)	9.83 (±4.24) ^a^	8.57 (±1.30)	8.17 (±0.70)	7.86 (±0.91)
Calcium (mg/L)	102.6 (±3.68)	97.5 (±7.23)	95.9 (±1.73)	99.2 (±2.22)	100.2 (±3.72)

Group of rats: C—control; Cd—exposed to cadmium; Cd + α-LA—exposed to cadmium and supplemented with α-lipoic acid; Cd + Mg—exposed to cadmium and supplemented with magnesium; Cd + Mg + α-LA—exposed to cadmium and supplemented with magnesium and α-lipoic acid). ^1^ Values are expressed as mean (±SD); ^a^ significant change from control (C), *p* < 0.05; ^b^ group exposed to Cd separately significantly different from groups exposed to Cd and supplemented with Mg and/or α-lipoic (Cd vs. Cd + α-LA or Cd vs. Cd + Mg or Cd vs. Cd + Mg + α-LA), *p* < 0.05.

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
