# Peer review of "Alleviating Effect of α-Lipoic Acid and Magnesium on Cadmium-Induced Inflammatory Processes, Oxidative Stress and Bone Metabolism Disorders in Wistar Rats"

_ijerph, 2019, doi:10.3390/ijerph16224483_

Round 1

Reviewer 1 Report

The manuscript entitled 2Alleviating effect of alpha-lipoic acid and magnesium on cadmium-induced inflammatory processes, oxidative stress and bone metabolism disorders in Wistar rats” submitted to International Journal of Environmental Research and Public Health” by Dr.Markiewics-Gorka and co-workers describes the effects on cadmium administration on blood, biochemical parameters, inflammatory and oxidative stress markers and bone metabolism. Furthermore, this study analyses the preventive effects of alpha-lipoic acid and magnesium on cadmium-induced toxicity. Obviously the study has interest but in my opinion, some aspects should be revised before to be published.

Line 24 and line 33. Inflammation should be changed by inflammatory markers. Authors did not use inflammation models. They measured some inflammatory marker such as protein C reactive. In my opinion the study did not present findings to support an anti-inflammatory effect of treatments as appoint authors along the manuscript (ex. line 829, line 947 “Mg and lipoic acid alleviate inflammation”, line 963 “anti-inflammatory”). Line 956. “reduction inflammation” How was measured this reduction of inflammation?

Line 34 “…alleviate ……… liver and kidney function” I can not understand this sentence. The manuscript needs a global English revision.

Line 126. What was chosen Cd concentration? Is a frequent exposure? What were chosen Mg and lipoic acid concentrations? Lipoic acid was orally administrated using syringe. Why? Lipoic could be included to diet. This orally administration can be stressed and consequently modify the findings of this group.

Line 140. Why a liquid diet?

Line 188. Authors measured thiobarbituric acid reactive substances (TBARs) that include MDA. However, MDA determination needs a chromatographic technique. In my opinion MDA should be changed by TBARs along the manuscript.

Asterisk and letters are used to appoint statistical significance (figures and tables). Why? Perhaps only letters is enough.

The main results of Tables 9, 19 and 11 should be included in results and/or discussion section. Thus these tables could be deleted.

Line 632. I can not understand the expression “nutritional status” (line 958 “body nutrition”…) in the context of this study. References 20-22 did not support this idea.

Line 822. Some sentences are too much ambiguous. “Magnesium may reverse these unfavorable processes”

Line 855. What means “magnesium antagonist”

Line 860. “…Mg is released from bone and consequently its serum level increases” Where are the findings on Mg bone release?

Line 922. Authors did not show that “cadmium increases permeability of cell membranes”

Author Response

Response to Reviewer 1 Comments

Dear Reviewer,

thank you very much for your helpful review, additional comments, and suggestions. We have revised our paper in line with the comments below.

Yours sincerely,

Iwona Markiewicz-Górka

Point 1: Line 24 and line 33. Inflammation should be changed by inflammatory markers. Authors did not use inflammation models. They measured some inflammatory marker such as protein C reactive. In my opinion the study did not present findings to support an anti-inflammatory effect of treatments as appoint authors along the manuscript (ex. line 829, line 947 “Mg and lipoic acid alleviate inflammation”, line 963 “anti-inflammatory”). Line 956. “reduction inflammation” How was measured this reduction of inflammation?

Response 1:

Thank you for pointing out this incorrect term. This has already been corrected throughout the text.

We considered the decrease in CRP and leukocyte count as a reduction in inflammation. However, I agree with the Reviewer that this was an unfortunate wording and over-interpretation. These errors have already been corrected in the text.

Point 2: Line 34 “…alleviate ……… liver and kidney function” I can not understand this sentence. The manuscript needs a global English revision.

Response 2:

Thank you for pointing out this illogical sentence. It has been replaced with the following statement:

“The obtained results show that both magnesium and α-lipoic acid decrease oxidative stress and the levels of inflammatory marker, as well as normalize bone metabolism and liver and kidney function”.

The manuscript has been checked and improved in terms of clarity and correctness of the English language.

Point 3: Line 126. What was chosen Cd concentration? Is a frequent exposure? What were chosen Mg and lipoic acid concentrations?

Response 3:

Thank you for this remark. The available literature indicates that a continuous exposure of rats to cadmium concentrations in the range of 5–50 mg Cd/l of drinking water reflects the level of human exposure to this metal: from moderate to relatively high (e.g. cigarette smokers, professionals exposed to Cd). In our experiment, we administered cadmium to rats with liquid feed and applied an intermediate dose (30 mg Cd/l), with the aim of inducing adverse biological effects, but not very severe pathological changes. In our previous experiment, the dose of 20 mg Cd/l administered with liquid diet for 5 months induced changes in bone morphology and metabolism, as well as disorders in hematological parameters and organ functioning (data not published). In the presented study, in order to induce similar changes but within a shorter time (3 months), we decided on a slightly higher dose, but remaining within the scope of occupational and high environmental exposures. Literature data report that α-lipoic acid administered to rats at the doses of 25–100 mg/kg body weight increases GSH levels and protects kidneys from cisplatin-related damage. Intraperitoneal administration of α-lipoic acid to older rats at a dose of 100 mg/kg body weight for 7 and 14 days increased the level of antioxidants (vitamin C, vitamin E, and reduced glutathione) and reduced the level of lipid peroxidation in the brain. In order to reduce the animals’ stress, we administered α-lipoic acid orally, and the applied dose was comparable with the therapeutic doses used in the above-mentioned studies and equaled 100 mg/kg body weight, 4 times per week. Magnesium was added to liquid feed at a dose of 150 mg Mg/l, which corresponds to the content of this element in good mineral waters available on the market (recommended in health prevention). The recommended dose of Mg for adult men equals 400–420 mg daily (6 mg/kg body weight). Magnesium is, however, relatively low in toxicity, and higher doses can be used for specific medical problems. In our experiment, the dose of magnesium estimated on the basis of the amount of feed consumed by the animals was 38 mg Mg/kg body weight and was higher than that recommended for healthy people. However, we used it for therapeutic purposes. Moreover, it can be assumed that the intake of Mg with feed containing Cd reduced its absorption from the gastrointestinal tract. The rationale for the doses has already been described in the article.

Point 4: Lipoic acid was orally administrated using syringe. Why? Lipoic could be included to diet. This orally administration can be stressed and consequently modify the findings of this group.

Response 4:

In our experience, we used a high purity, relatively expensive reagent. The administration of α-lipoic acid with feed that was freshly prepared every other day, with leftovers poured out, would be uneconomical. The caretakers in our laboratory love animals. The rats became accustomed very quickly, let people stroke them, and did not run away from an open cage. For the animals, the oral administration of α-lipoic acid with a syringe was just an additional contact with their favorite caretaker (stroking, holding, caressing words) rather than significant stress that could modify the results. The biggest stress was that of the caretaker at the end of the experiment.

Point 5: Line 140. Why a liquid diet?

Response 5:

We administered cadmium with feed because this reflects environmental exposure better than Cd administration with drinking water. People also consume more cadmium with their food than with water. In addition, rats do not drink too much water, so a higher Cd concentration would have to be used to obtain similar biological effects.

Liquid diet makes it much easier to monitor the volume of feed consumed by animals and the cadmium intake. The rats had special wooden blocks in their cages to grind their teeth.

Point 6: Line 188. Authors measured thiobarbituric acid reactive substances (TBARs) that include MDA. However, MDA determination needs a chromatographic technique. In my opinion MDA should be changed by TBARs along the manuscript.

Response 6:

Thank you for pointing out these inaccuracies. The issue has been corrected in the text as suggested by the Reviewer.

Point 7: Asterisk and letters are used to appoint statistical significance (figures and tables). Why? Perhaps only letters is enough.

Response 7:

The asterisk was used to improve the clarity of the figures and tables, i.e. to distinguish the changes in relation to the control group from the differences between the individual cadmium-poisoned groups. However, the Reviewer found that less readable, therefore all the symbols of statistically significant differences have been replaced by letters.

Point 8: The main results of Tables 9, 19 and 11 should be included in results and/or discussion section. Thus these tables could be deleted.

Response 8:

Thank you for the suggestion. The tables have been removed and statistically significant correlations are now presented in a descriptive way in the results section.

Point 9: Line 632. I can not understand the expression “nutritional status” (line 958 “body nutrition”…) in the context of this study. References 20-22 did not support this idea.

Response 9:

Thank you for this remark. I agree with the Reviewer that the terms “nutritional status” and “body nutrition” in the sentences referred to above were not properly used. They have been removed from the text. In our experiment, only lower body weight gain in rats poisoned with cadmium indicated their poorer nutritional status. A similar approach to other changes, i.e. those in hematological parameters or decrease in cholesterol concentration, was a mistake and over-interpretation.

Point 10: Line 822. Some sentences are too much ambiguous. “Magnesium may reverse these unfavorable processes”

Response 10:

Thank you for this remark. The sentence has been removed from the text.

Point 11: Line 855. What means “magnesium antagonist”

Response 11:

Thank you for this remark. We used this term quite unsuitably to describe the phenomenon of Cd and Mg competing in the body, among others on the level of absorption, binding with proteins transporting metals in the gastrointestinal tract, and accumulation in tissues. The sentence has been changed and the wrong term has been removed from the text.

Point 12: Line 860. “…Mg is released from bone and consequently its serum level increases” Where are the findings on Mg bone release?

Response 12:

Thank you for this remark. These are only our assumptions based on the growth of magnesium levels in the serum of animals from the group intoxicated with Cd only, as well as on literature data. As the Reviewer rightly pointed out, we did not document this phenomenon in our experiment. The statement has been removed from the text.

Point 13: Line 922. Authors did not show that “cadmium increases permeability of cell membranes”

Response 13:

Thank you for this remark. Increased activity of liver enzymes in serum indicates damage to liver cells and raised permeability of cell membranes. However, this sentence was in fact an over-interpretation and has been removed from the text.

Reviewer 2 Report

The authors sought to determine the effect of a–lipoic acid and magnesium exposure on Cd-induced toxicity in Wistar rats. Given the worldwide abundance of Cd, and increasing human Cd exposure, research on the underlying mechanisms in neurotoxicity, and how to counteract this toxicity is worth investigating.

The work is well planned and performed. The conclusions are supported by the experimental evidence. However, I have some minor concerns affecting mainly to the way in which results are presented that I would like to see addressed by the authors before publication.

ABSTRACT

Please, consider to introduce in the abstract the concentrations employed in the experiments for each of the agents.

Although the manuscript provides an overall well-written piece of work, I have one suggestion for further improvement. The manuscript is at some points pretty hard to read. This is mainly due to the sometimes quite complex structure of the sentences as well as the choice of wording (see e.g. results description). I would advise that the authors take some time to re-read the manuscript and reduce the complexity of sentences where possible, improving the readability.

Author Response

Response to Reviewer 2 Comments

Dear Reviewer,

thank you very much for your helpful review. We have revised our paper in line with the comments below.

Yours sincerely,

Iwona Markiewicz-Górka

Point 1: ABSTRACT

Please, consider to introduce in the abstract the concentrations employed in the experiments for each of the agents.

Response 1:

Thank you for the suggestion. It has been included in the text.

Point 2: Although the manuscript provides an overall well-written piece of work, I have one suggestion for further improvement. The manuscript is at some points pretty hard to read. This is mainly due to the sometimes quite complex structure of the sentences as well as the choice of wording (see e.g. results description). I would advise that the authors take some time to re-read the manuscript and reduce the complexity of sentences where possible, improving the readability.

Response 2:

Thank you for the suggestion. The text has been corrected and simplified. We hope that now it will be more readable.

Reviewer 3 Report

The submitted article title: Alleviating Effect of α-Lipoic Acid and Magnesium on Cadmium-Induced Inflammatory Processes, Oxidative Stress and Bone Metabolism Disorders in Wistar Rats

Summary: This is paper describes a number of mechanism based blood and selected tissue sample measurements collected from a study of rats treated 3 months to cadmium or co-treated to cadmium plus magnesium and/or α-lipoic acid.  Although mostly a descriptive paper the authors proved an adequate discussion on the potential impact of the findings.  I recommend acceptance with several minor revisions.

Did the authors collect in life observations, organ weights and gross observations of the liver and kidney during necropsy or follow-up histology of the collected organs? That information would be very useful and should be included as supplemental information. If these measurements were not collected, a reason or at least a mention should be provided in the methods section. It appears from the provided references of human cadmium exposure to suggest there are no significant differences between men and women. The authors cite studies looking at men and women for the inflammation marker CRP but none of the other measurements except DPD and BMD in post-menopausal women. Is there any human exposure data available for the other measurements of this study? If there is, it should be included and if not, this should be mentioned.   Table 3. In the legend the superscript ‘a’ is orphaned from Groups. These need to be kept together.

The first sentence of the conclusion is a little misleading. No histopathology is provided to suggest ‘toxic’ effects were weakened with magnesium and/ or α-lipoic acid. This is particularly important as the authors discuss a study finding structural and functional damage to proximal tubules and mitochondrial. This sentence should be clarified to read: ‘Despite the relatively higher doses of cadmium consumed with feed by animals supplemented with magnesium and/or α-lipoic acid, the toxic effects measured in this study were weakened compared with the changes observed in rats receiving only cadmium in feed.

Author Response

Response to Reviewer 3 Comments

The submitted article title: Alleviating Effect of α-Lipoic Acid and Magnesium on Cadmium-Induced Inflammatory Processes, Oxidative Stress and Bone Metabolism Disorders in Wistar Rats

Summary: This is paper describes a number of mechanism based blood and selected tissue sample measurements collected from a study of rats treated 3 months to cadmium or co-treated to cadmium plus magnesium and/or α-lipoic acid. Although mostly a descriptive paper the authors proved an adequate discussion on the potential impact of the findings. I recommend acceptance with several minor revisions.

Dear Reviewer,

thank you very much for your helpful review, additional comments, and suggestions. We have revised our paper in line with the comments below.

Yours sincerely,

Iwona Markiewicz-Górka

Point 1: Did the authors collect in life observations, organ weights and gross observations of the liver and kidney during necropsy or follow-up histology of the collected organs? That information would be very useful and should be included as supplemental information. If these measurements were not collected, a reason or at least a mention should be provided in the methods section.

Response 1:

Thank you for the remark. The rats were under constant observation of the laboratory workers. No worrying signs in behavior, activity, or appearance were observed during the experiment. No worrying clinical signs were found such as diarrhea, hair bristling, cage soaking, pale eyes, or distended abdomen (the observation forms were filled in daily).

During the autopsy, a macroscopic evaluation of the organs was performed. No changes were observed in the shape, size, color, or consistency of liver or kidneys in the cadmium-poisoned groups as compared with the organs in the control group. The livers and kidneys were weighed. Relative mass was calculated for liver and kidney. Moreover, absolute mass was compared for kidneys (as more reliable than RKM for identifying potential renal toxicants)1.

The differences in the absolute mass of kidneys across the rat groups were not statistically significant. Higher RLM and RKM was observed in Cd groups supplemented with Mg and/or α-lipoic in relation to the control group. Among all groups of rats, the highest relative liver and kidney mass was reported in groups supplemented with α-lipoic acid separately. However, owing to lower values of liver enzymes activity and better biochemical parameters (in comparison to the Cd group), these changes were considered as an adaptive response of the organ to increased functional load associated with the metabolism of xenobiotics and supplement. As suggested by the Reviewer, these results have been included as supplementary information.

Histopathological evaluation was not performed because of the high number of biochemical tests, as well as for economic and organizational reasons.

1Craig E.A. et al. The relationship between chemical-induced kidney weight increases and kidney histopathology in rats J. Appl. Toxicol. 2015; 35: 729–736.)

Point 2: It appears from the provided references of human cadmium exposure to suggest there are no significant differences between men and women. The authors cite studies looking at men and women for the inflammation marker CRP but none of the other measurements except DPD and BMD in post-menopausal women. Is there any human exposure data available for the other measurements of this study? If there is, it should be included and if not, this should be mentioned.

Response 2:

Thank you for the suggestion. In the mentioned population-based study of Swedish men and women, the aim was to investigate the relationship between blood cadmium (B-Cd) at baseline and cardiovascular incidents in a 17-year follow-up. The authors observed that hazard ratios for all cardiovascular incidents were higher in patients within the fourth B-Cd quartile. The least number of women were in quartile I. In addition to higher levels of CRP, quartile IV was characterized by higher average levels of glycated hemoglobin (HbA1c), a higher rate of smokers, and higher prevalence of low education and low physical activity. For other evaluated cardiovascular risk factors, such as LDL, HDL, triglycerides, diabetes, blood pressure, the differences among B-Cd quartiles were small.

The authors suggest that in terms of lowering the frequency of cardiovascular incidents, it is reasonable to take measures to reduce exposure to cadmium, even in populations with moderate exposure.

The remarks have been included in the text.

Point 3: Table 3. In the legend the superscript ‘a’ is orphaned from Groups. These need to be kept together.

Response 3:

Thank you for pointing out this mistake; it has been corrected. As suggested by another Reviewer, the symbols in the legend have been changed to letters alone to increase readability. The previous orphaned “a” is now “b”.

Point 4: The first sentence of the conclusion is a little misleading. No histopathology is provided to suggest ‘toxic’ effects were weakened with magnesium and/ or α-lipoic acid. This is particularly important as the authors discuss a study finding structural and functional damage to proximal tubules and mitochondrial. This sentence should be clarified to read: ‘Despite the relatively higher doses of cadmium consumed with feed by animals supplemented with magnesium and/or α-lipoic acid, the toxic effects measured in this study were weakened compared with the changes observed in rats receiving only cadmium in feed.

Response 4:

Thank you for pointing out this inaccuracy. The sentence has been corrected as suggested by the Reviewer.

Round 2

Reviewer 1 Report

This last version of the manuscript considers the comments and suggestions proposed by the reviewer. Thus, I believe that this version could be considered to publish by International Journal of Environmental Research and Public Health